# A feedback loop between the androgen receptor and 6-phosphogluoconate dehydrogenase (6PGD) drives prostate cancer growth

Joanna L Gillis[1,2], Josephine A Hinneh[1,2,3], Natalie K Ryan[1,2], Swati Irani[1,2], Max Moldovan[2], Lake-Ee Quek[4], Raj K Shrestha[1,5,6,7], Adrienne R Hanson[5], Jianling Xie[5], Andrew J Hoy[8], Jeff Holst[9], Margaret M Centenera[1,2,7], Ian G Mills[10,11], David J Lynn[2,5], Luke A Selth[1,5,6,7]*, Lisa M Butler[1,2,7]*

[1]Adelaide Medical School, University of Adelaide, Adelaide, Australia; [2]South Australian Health and Medical Research Institute, Adelaide, Australia; [3]Department of Urology, Nagoya University Graduate School of Medicine, Nagoya, Japan; [4]School of Mathematics and Statistics, Charles Perkins Centre, Faculty of Science, The University of Sydney, Camperdown, Australia; [5]Flinders Health and Medical Research Institute, Flinders University, College of Medicine and Public Health, Bedford Park, Australia; [6]Dame Roma Mitchell Cancer Research Laboratories, University of Adelaide, Adelaide, Australia; [7]Freemasons Centre for Male Health and Wellbeing, University of Adelaide, Adelaide, Australia; [8]School of Medical Sciences, Charles Perkins Centre, Faculty of Medicine and Health, The University of Sydney, Camperdown, Australia; [9]School of Medical Sciences and Prince of Wales Clinical School, University of New South Wales, Sydney, Australia; [10]Centre for Cancer Research and Cell Biology, Queen's University Belfast, Northern Ireland, United Kingdom; [11]Nuffield Department of Surgical Sciences, University of Oxford, Oxford, United Kingdom

*For correspondence:
luke.selth@flinders.edu.au (LAS);
lisa.butler@adelaide.edu.au
(LMB)

**Competing interest:** The authors declare that no competing interests exist.

**Abstract** Alterations to the androgen receptor (AR) signalling axis and cellular metabolism are hallmarks of prostate cancer. This study provides insight into both hallmarks by uncovering a novel link between AR and the pentose phosphate pathway (PPP). Specifically, we identify 6-phosphogluoconate dehydrogenase (*6PGD*) as an androgen-regulated gene that is upregulated in prostate cancer. AR increased the expression of *6PGD* indirectly via activation of sterol regulatory element binding protein 1 (SREBP1). Accordingly, loss of 6PGD, AR or SREBP1 resulted in suppression of PPP activity as revealed by 1,2-$^{13}C_2$ glucose metabolic flux analysis. Knockdown of 6PGD also impaired growth and elicited death of prostate cancer cells, at least in part due to increased oxidative stress. We investigated the therapeutic potential of targeting 6PGD using two specific inhibitors, physcion and S3, and observed substantial anti-cancer activity in multiple models of prostate cancer, including aggressive, therapy-resistant models of castration-resistant disease as well as prospectively collected patient-derived tumour explants. Targeting of 6PGD was associated with two important tumour-suppressive mechanisms: first, increased activity of the AMP-activated protein kinase (AMPK), which repressed anabolic growth-promoting pathways regulated by acetyl-CoA carboxylase 1 (ACC1) and mammalian target of rapamycin complex 1 (mTORC1); and second, enhanced AR ubiquitylation, associated with a reduction in AR protein levels and activity. Supporting the biological relevance of positive feedback between AR and 6PGD, pharmacological co-targeting of both factors was more effective in suppressing the growth of prostate cancer cells than single-agent therapies. Collectively, this work provides new insight into the dysregulated metabolism of

prostate cancer and provides impetus for further investigation of co-targeting AR and the PPP as a novel therapeutic strategy.

## Introduction

Altered cellular metabolism is a hallmark of cancer. Perhaps the best characterised metabolic transformation in malignant cells is the so-called Warburg effect, in which cancer cells favour metabolism via glycolysis rather than the more efficient oxidative phosphorylation (*Vander Heiden et al., 2009*). While Warburg-like metabolism plays a key role in many malignancies, more recent work has demonstrated the diversity of cancer metabolism and revealed that tissue of origin is likely to be the critical determinant of malignant metabolic reprogramming (*Bader and McGuire, 2020*). One tissue that exhibits a unique metabolic profile is the prostate (*Lin et al., 2019*). Normal prostate epithelial cells exhibit a truncated tricarboxylic acid (TCA) cycle to enable production of citrate, a key component of prostatic fluid, resulting in high rates of glycolysis (*Bader and McGuire, 2020*). By contrast, malignant transformation switches metabolism of prostate cells to a more energetically favourable phenotype by re-establishing an intact TCA cycle, whereby citrate is utilised for oxidative phosphorylation and biosynthetic processes such as lipogenesis (*Flavin et al., 2011*).

A major regulator of the unique metabolism of the normal and malignant prostate is the androgen receptor (AR) (*Butler et al., 2016*). AR is a hormone (androgen)-activated transcription factor that regulates expression of a large suite of genes involved in various aspects of metabolism, either directly or indirectly through activation of other master regulators such as sterol regulatory element-binding protein-1 (SREBP1) (*Gonthier et al., 2019*; *Heemers et al., 2003*). Given its integral metabolic functions, it is unsurprising that AR is the primary oncogenic driver of prostate cancer (PCa) and the major therapeutic target in advanced and metastatic disease. While suppression of AR activity by androgen receptor pathway inhibitors (ARPIs) is initially effective in almost all men, prostate tumours inevitably develop resistance and progress to a lethal disease state known as castration-resistant prostate cancer (CRPC). One key feature of CRPC is the maintenance or reactivation of the AR signalling axis, as revealed by the therapeutic benefit of second-generation ARPIs, such as the AR antagonist enzalutamide, in CRPC (*Beer et al., 2014*). Unfortunately, the overall survival benefits of these newer ARPIs in men with CRPC are in the order of months (*Recine and Sternberg, 2015*), despite many tumours retaining dependence on AR (*Robinson et al., 2015*). Collectively, these clinical observations highlight the ongoing dependence of CRPC on AR signalling and the intractable problems associated with therapies that inhibit this pathway.

Direct alterations to AR – including mutation, amplification, alternative splicing, and altered ligand availability – have been well characterised as mechanisms of resistance in CRPC (*Coutinho et al., 2016*). However, the extent to which AR-mediated metabolic reprogramming is involved in therapy resistance in CRPC is less well understood. Herein, using an unbiased approach to discover potential PCa survival factors, we identify 6-phosphogluoconate dehydrogenase (6PGD) as a novel AR-regulated factor. 6PGD is a key enzyme in the pentose phosphate pathway (PPP) (also referred to as the phosphogluconate pathway or the hexose monophosphate shunt), an alternative metabolic pathway for glucose breakdown. The PPP comprises two phases: an irreversible oxidative phase that generates NADPH and ribulose 5-phosphate (Ru5P); and a subsequent reversible non-oxidative phase in which Ru5P is converted to ribose 5-phosphate (R5P), a sugar precursor for generation of nucleotides (*Jin and Zhou, 2019*; *Ge et al., 2020*). NADPH produced by the PPP is used for many anabolic reactions, including fatty acid synthesis, as well as an electron donor to generate reduced glutathione, the major endogenous antioxidant (*Ge et al., 2020*). Thus, the PPP is a major regulator of both redox homeostasis as well as anabolic reactions, depending on cellular requirements. We demonstrate that 6PGD plays a key role in PCa growth and survival, at least in part through moderating oxidative stress, and uncover a novel feedback mechanism linking 6PGD and the AR signalling axis that provides impetus for further investigation of co-targeting AR and the PPP as a novel therapeutic strategy.

## Results

### *6PGD* is an androgen-regulated gene in PCa

The current clinical ARPIs, such as enzalutamide, do not target the entire repertoire of genes regulated by the AR in prostate tumour cells (*Asangani et al., 2014*). We hypothesised that ablation of AR expression would be the most appropriate 'therapeutic benchmark' to identify the key regulators of tumour cell survival regulated by AR. To qualitatively and quantitatively compare downstream responses to AR ablation and AR antagonism, LNCaP cells were treated with AR siRNA (siAR; i.e. AR ablation) or enzalutamide (Enz; AR antagonism) and subsequently evaluated by RNA-seq. The experimental conditions were optimised to achieve comparable suppression of the canonical AR target, prostate specific antigen (PSA), which is encoded by the *KLK3* gene (*Figure 1A*). Genes affected by siAR were highly concordant with an independent dataset (*He et al., 2014*; *Figure 1—figure supplement 1A*). As expected, most (78%) genes altered by enzalutamide (compared to vehicle control) were also similarly dysregulated by siAR (compared to a control siRNA [siCon]) (*Figure 1B*, *Figure 1—source data 1*). An additional 2574 genes were altered in their expression by siAR but not enzalutamide (*Figure 1B*; q < 0.05). On closer examination, many of these genes were altered in their expression by enzalutamide but not sufficiently for them to be identified as statistically significant differentially expressed genes. A further direct statistical comparison of gene expression between the two treatment groups identified that there were 581 genes that were differentially expressed in the siAR-treated cells compared to those treated with enzalutamide including, as expected, *AR* itself (*Figure 1B,C*, *Figure 1—source data 1*; q < 0.05). These results provide further evidence for the hypothesis that AR ablation is more effective at suppressing the AR-regulated transcriptome compared with AR antagonism, at least in this experimental system.

The gene most significantly associated with AR ablation and not AR antagonism was *6PGD* (*Figure 1C*, *Figure 1—source data 1*), which encodes an enzyme in the PPP. We confirmed that 6PGD expression was downregulated by AR knockdown but not by acute AR antagonism in multiple PCa cell lines (LNCaP and VCaP) at both the mRNA and protein level (*Figure 1D,E*; *Figure 1—figure supplement 1B–D*). Downregulation of 6PGD was also seen with a second AR siRNA, validating 6PGD as a bona fide target of AR (*Figure 1—figure supplement 1B–D*). In further support of differential regulation by siAR versus AR antagonism, neither of the newest clinically approved AR antagonists (apalutamide and darolutamide) altered 6PGD protein or mRNA expression (*Figure 1—figure supplement 1E,F*). Conversely, AR activation with the androgen 5α-dihydrotestosterone (DHT) stimulated *6PGD* expression, and this effect was abolished by co-treatment with siAR (*Figure 1F*). To determine whether AR inhibition affects 6PGD in more biologically relevant systems, we utilised our patient-derived explant (PDE) model (*Centenera et al., 2018*). Similar to two-dimensional PCa cell line culture, we did not observe enzalutamide-mediated changes to *6PGD* mRNA expression in the PDE model over a time frame of 48 hr under conditions that caused significant repression of the well-characterised AR target genes *KLK2* and *KLK3* (*Figure 1G*). By contrast, longer-term (~14 weeks) androgen deprivation therapy in patients caused a significant decrease in *6PGD* mRNA levels (*Figure 1H*). Collectively, these findings reveal 6PGD as a novel AR-regulated factor in both PCa cell lines and clinical samples.

As an initial assessment of the relevance of 6PGD in clinical PCa, we examined its expression in a clinical transcriptomic dataset (*Cancer Genome Atlas Research Network, 2015*) and found that *6PGD* mRNA expression was significantly elevated in cancer compared to patient-matched normal tissue and also showed an association with increasing Gleason grade (*Figure 1I,J*), although it was not associated with biochemical recurrence (data not shown). An association with malignancy was recapitulated at the protein level (*Figure 1K*) in a distinct set of patient samples for which proteomes were profiled using mass spectrometry (*Latonen et al., 2018*). We further examined 6PGD protein expression in prostate tumours by immunohistochemistry (IHC). 6PGD was detected in all tissues that were examined and was predominantly localised to the cytoplasm and perinuclear regions of epithelial cells (*Figure 1—figure supplement 2*). Moreover, we observed a trend towards increasing protein levels in the more aggressive tumours (*Figure 1—figure supplement 2*). In summary, 6PGD is highly expressed in prostate tumours, suggesting that the PPP may play an important metabolic role in this cancer type.

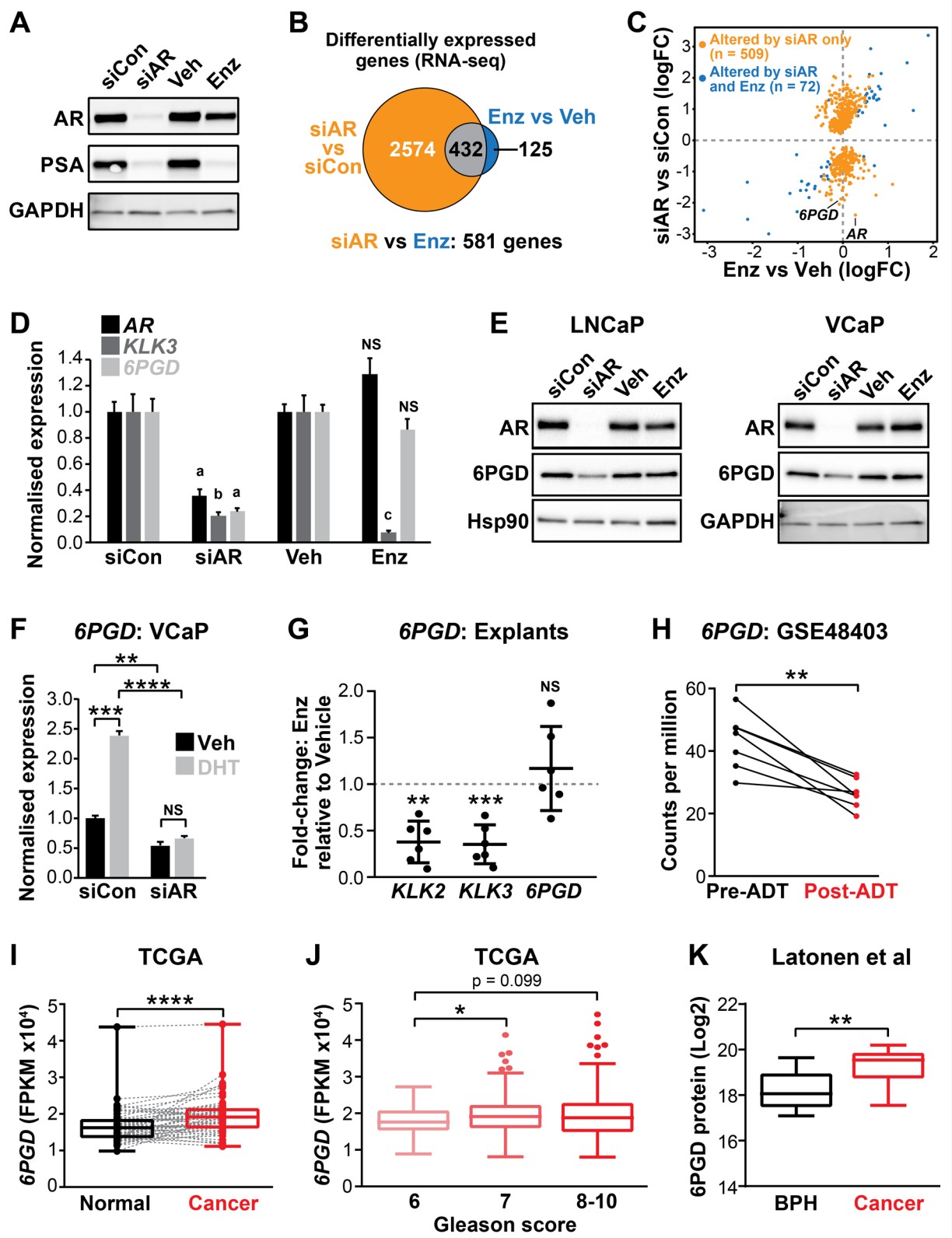

**Figure 1.** *6PGD* is an androgen receptor (AR)-regulated gene and is elevated in prostate cancer. (**A**) Effect of siAR and enzalutamide (Enz) on the AR target, PSA. LNCaP cells were transfected with AR (siAR; 12.5 nM) or control (siCon) siRNA for 48 hr or treated with Enz (1 μM) or vehicle (Veh) for 24 hr, after which AR and PSA proteins were evaluated by immunoblotting. GAPDH was used as loading control. (**B**) Numbers of genes differentially expressed (false discovery rate [FDR] < 0.05) by siAR (versus siCon) or Enz (vs. Veh) are shown in the Venn diagram (at top). Below: an alternative analysis identified

*Figure 1 continued on next page*

*Figure 1 continued*

581 genes differentially expressed (FDR < 0.05) by siAR versus Enz. (**C**) Scatterplot of genes affected by siAR and Enz. The 581 genes differentially expressed by siAR versus Enz are shown in blue (n = 72, genes differentially expressed by siAR versus siCon and Enz versus Veh) and yellow (n = 509), genes differentially expressed by siAR versus siCon but not by Enz versus Veh. (**D**) Validation of *6PGD* expression in response to siAR and Enz by RT-qPCR. Gene expression was normalised to *GUSB* and *L19* and represents the mean ± standard error of the mean (SEM) of three biological replicates; siCon and Veh were set to 1. Differential expression was evaluated using unpaired t tests (a, p<0.01; b, p<0.001; c, p<0.0001; NS, not significant). (**E**) 6PGD protein levels in response to siAR and Enz treatments were measured by immunoblotting in LNCaP (left) and VCaP (right) cells. HSP90 and GAPDH were used as loading controls. (**F**) RT-qPCR of *6PGD* expression in response to DHT and siAR in VCaP cells. Cells were transfected with siRNAs for 24 hr, and then treated with 1 nM DHT for another 24 hr. Gene expression was normalised and graphed as in (**D**). Differential expression was evaluated by t tests (**p < 0.01; ***p < 0.001; ****p < 0.0001). (**G**) RT-qPCR of *KLK2, KLK3,* and *6PGD* expression in response to Enz treatment (1 µM, 72 hr) in patient-derived explants. Gene expression was normalised to *GAPDH, PPIA,* and *TUBA1B* and is represented as fold-change relative to vehicle. Differential expression was evaluated by one-sample t tests (**p<0.01; ***p<0.001). (**H**) *6PGD* mRNA expression in prostate tumours pre- and post-androgen deprivation therapy (ADT; GSE48403). A Wilcoxon matched-pairs signed-rank test was used to compare expression in the groups. (**I**) *6PGD* expression is elevated in primary prostate cancer. The TCGA dataset comprises 52 patient-matched normal and cancer samples. Boxes show minimum and maximum (bottom and top lines, respectively) and mean (line within the boxes) values. A paired t test was used to compare expression in normal versus cancer. FPKM: fragments per kilobase of exon per million mapped reads. (**J**) *6PGD* expression by Gleason grade in the TCGA cohort. Boxes show minimum and maximum (bottom and top lines, respectively) and mean (line within the boxes) values. Unpaired t tests were used to compare expression between the groups. (**K**) 6PGD protein expression in clinical prostate samples (benign prostatic hyperplasia [BPH] and tumours) was measured mass spectrometry. Boxes show minimum and maximum (bottom and top lines, respectively) and mean (line within the boxes) values. An unpaired t test was used to compare expression between the groups.

The online version of this article includes the following source data and figure supplement(s) for figure 1:

**Source data 1.** Differentially-expressed genes in LNCaP prostate cancer cells treated with siAR or Enz.

**Figure supplement 1.** 6PGD expression is decreased by AR knockdown but not by AR inhibition.

**Figure supplement 2.** Representative images of 6PGD immunohistochemistry in patient tumours.

## SREBP1 mediates induction of *6PGD* downstream of the AR

AR binds to gene enhancers or promoters to directly regulate transcription (*Wang et al., 2007*). However, we found no clear evidence of AR binding sites proximal to the *6PGD* transcriptional start site in genome-wide DNA binding (ChIP-seq) datasets from tissues and cell lines (*Figure 2A* and data not shown), suggesting that the AR pathway may indirectly regulate *6PGD* expression via another downstream pathway(s) or factor(s). One credible intermediary between AR and 6PGD is SREBP1, a master transcriptional regulator of genes involved in lipid and cholesterol production (*Heemers et al., 2006*). AR enhances SREBP1 expression and activity in a multifaceted manner, most notably by upregulating the SREBP1 activator SCAP (*Heemers et al., 2006*) and by activating the mTOR pathway, which in turn leads to elevated SREBP1 expression (*Duvel et al., 2010*). Additionally, SREBP1 has been proposed to directly regulate *6PGD* in mouse adipocytes by direct binding to its promoter (*Rho et al., 2005*). We mined ENCODE SREBP1 ChIP-seq data and identified an SREBP1 binding site at the *6PGD* promoter in two cancer cell lines, HEPG2 (liver) and MCF7 (breast) (*Figure 2B*). Regulation of 6PGD by SREBP1 in PCa cells was confirmed by siRNA-mediated knockdown of SREBP1 (*Figure 2C*). To test whether SREBP1 acts downstream of AR to increase *6PGD* expression, we treated LNCaP cells with siSREBP1 or a pharmacological inhibitor of SREBP1 (fatostatin) and then evaluated 6PGD expression in the presence or absence of DHT. Supporting our hypothesis, either knockdown (*Figure 2D*) or inhibition of SREBP1 antagonised androgen-mediated induction of 6PGD (*Figure 2E*). We validated this effect in an independent AR-responsive cell line, VCaP (*Figure 2D*). Collectively, these results reveal the presence of a functional AR-SREBP1-6PGD circuit in PCa cells and implicate SREBP1 as a key mediator of PPP activation by AR.

## An AR-SREBP1-6PGD axis influences PCa cell growth and activity of the pentose phosphate pathway

Regulation of 6PGD by the AR signalling axis supports other recent reports linking the PPP to PCa (*Tsouko et al., 2014*; *Ros et al., 2012*); and although the role of the PPP in this malignancy is not fully elucidated, it could serve to fuel cell growth and protect against oxidative stress. In support of this, knockdown of 6PGD with two highly effective siRNAs (*Figure 3—figure supplement 1*) significantly decreased viability (*Figure 3A*) and increased death (*Figure 3B*) of LNCaP and VCaP cells. These findings were recapitulated in cell line models of CRPC (V16D) and enzalutamide-resistant CRPC (MR49F)

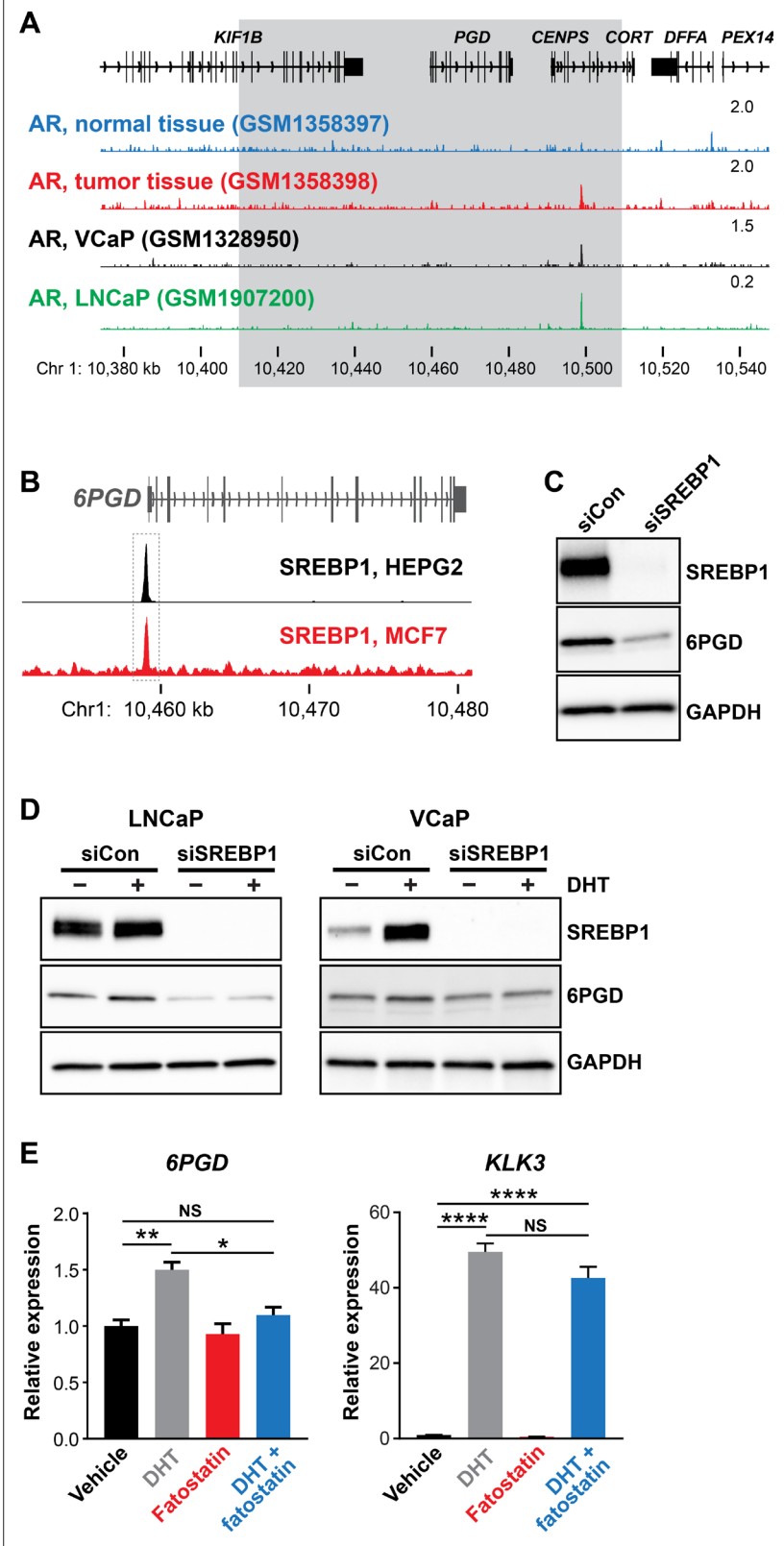

**Figure 2.** SREBP1 mediates induction of 6PGD downstream of the androgen receptor (AR). (**A**) ChIP-seq data showing AR DNA binding near the *6PGD* gene in non-malignant and prostate tumour samples (***Pomerantz et al., 2015***) and the LNCaP (***Barfeld et al., 2017***) and VCaP (***Asangani et al., 2014***) cell line models. The grey box indicates a region ±50 kb of the *6PGD* transcriptional start site. (**B**) ChIP-seq data showing SREBP1 DNA

*Figure 2 continued on next page*

*Figure 2 continued*

binding at the *6PGD* promoter in HEPG2 and MCF7 cells. Data is from ENCODE (***ENCODE Project Consortium, 2012***; HEPG2: ENCFF000XXR; MCF7: ENCFF911YFI). (**C**) Effect of siSREBP1 on 6PGD protein. LNCaP cells were transfected with siRNA (siSREBP1; 12.5 nM) or control (siCon) for 72 hr after which SREBP1 and 6PGD protein levels were evaluated by immunoblotting. GAPDH was used as loading control. (**D**) Effect of siSREBP1 on 6PGD induction by DHT. LNCaP (left) or VCaP (right) cells were transfected with siRNA (siSREBP1; 12.5 nM) or control (siCon) in charcoal-stripped FBS media for 72 hr and then treated with 10 nM DHT for another 24 hr. SREBP1 and 6PGD protein levels were evaluated by immunoblotting. GAPDH was used as loading control. (**E**) RT-qPCR of *6PGD* expression in response to DHT and fatostatin in LNCaP cells. Cells were serum starved in charcoal-stripped FBS media for 72 hr and then treated with Veh or 10 nM DHT ±10 μM fatostatin for another 24 hr. Gene expression was normalised to *GUSB* and *L19* and represents the mean + SEM of three biological replicates. Treatment effects were evaluated using ANOVA and Dunnett's multiple comparison tests (*p<0.05; **p<0.01; ****p<0.0001; NS, not significant).

(***Figure 3A,B***). In addition to these phenotypic effects, mass spectrometry revealed accumulation of 6PGD's substrate, 6-phosphogluconate (6- PG) (***Figure 3C***), in LNCaP cells transfected with siRNA, confirming specificity of the knockdown.

To more directly investigate the involvement of 6PGD, AR, and SREBP1 in the PPP, we conducted mass spectrometry tracing experiments with 1,2-$^{13}$C$_2$ glucose. After 48 hr of siRNA transfection, 1,2-$^{13}$C$_2$ glucose was spiked in to growth media at a ratio of 1:1 with natural glucose and PPP flux was estimated over a period of 15 min by measuring the incorporation of $^{13}$C into the immediate product of 6PGD's catalytic activity, ribulose 5-phosphate (Ru5P). A schematic detailing the differential incorporation of $^{13}$C isotope into Ru5P by both the oxidative (irreversible; m1 Ru5P) and non-oxidative (reversible; m2 Ru5P) branches of the PPP is shown in ***Figure 3D***. Isotopic steady-state enrichments of glucose 6-phosphate (G6P) confirmed that approximately 1:1 ratio labelling was achieved consistently between treatment groups (***Figure 3E***), demonstrating that PPP flux could be inferred from labelled Ru5P without correcting for enrichment bias between treatments. Next, we used the accumulation profiles of m1 (singly labelled) Ru5P (***Figure 3F***) to estimate the rate of Ru5P production via 6PGD from exogenous glucose (i.e. dilution rate; ***Figure 3G***). These analyses revealed that flux through the oxidative PPP was significantly decreased with knockdown of 6PGD, AR, and SREBP1 (***Figure 3F,G***). Interestingly, knockdown of AR and SREBP1 (but not 6PGD) also had a significant impact on flux through the non-oxidative phase of the PPP, as determined by evaluating m2 (doubly labelled) Ru5P production via F6P/GAP (***Figure 3F,G***). Collectively, these glucose tracing data show that targeting 6PGD significantly suppresses PPP activity through the oxidative pathway, an effect that is also evident when targeting the upstream signalling factors AR and SREBP1.

Since a key role of the PPP is to regulate intracellular redox state (***Ge et al., 2020***), we also measured reactive oxygen species (ROS) using a flow cytometric-based assay. Knockdown of 6PGD (and AR) significantly increased levels of intracellular ROS in both androgen-sensitive and CRPC cell line models (***Figure 3G***). This phenotype could be rescued by the antioxidant Trolox (***Figure 3H***), verifying the specificity of the assay.

## Pharmacological inhibition of 6PGD suppresses PCa growth and increases ROS

Having established that 6PGD is required for efficient activity of the PPP, optimal PCa cell growth and protection against oxidative stress, we evaluated pharmacological targeting of this enzyme as a potential therapeutic strategy. Physcion, a plant-derived anthraquinone, was recently identified as an inhibitor of 6PGD using an in vitro screening assay (***Lin et al., 2015***). Treatment of LNCaP cells with physcion dose-dependently inhibited growth and elicited death (***Figure 4—figure supplement 1A,B***). However, low solubility limits the preclinical and clinical utility of this compound. Therefore, we focused our efforts on a derivative of physcion, S3, which has substantially improved solubility (~50-fold: 1 mM physcion c.f. 50 mM S3 in DMSO; ***Lin et al., 2015***). Similarly to physcion, S3 reduced LNCaP cell viability and caused cell death (***Figure 4A,B***). Cell kill was at least partly mediated via apoptosis, as demonstrated by a flow cytometric-based Annexin/7-AAD assay (***Figure 4C***). Importantly, S3 increased levels of cellular ROS in a dose-dependent manner (***Figure 4D***), strengthening the link between the PPP and control of redox homeostasis. S3 was active in a range of PCa models, including

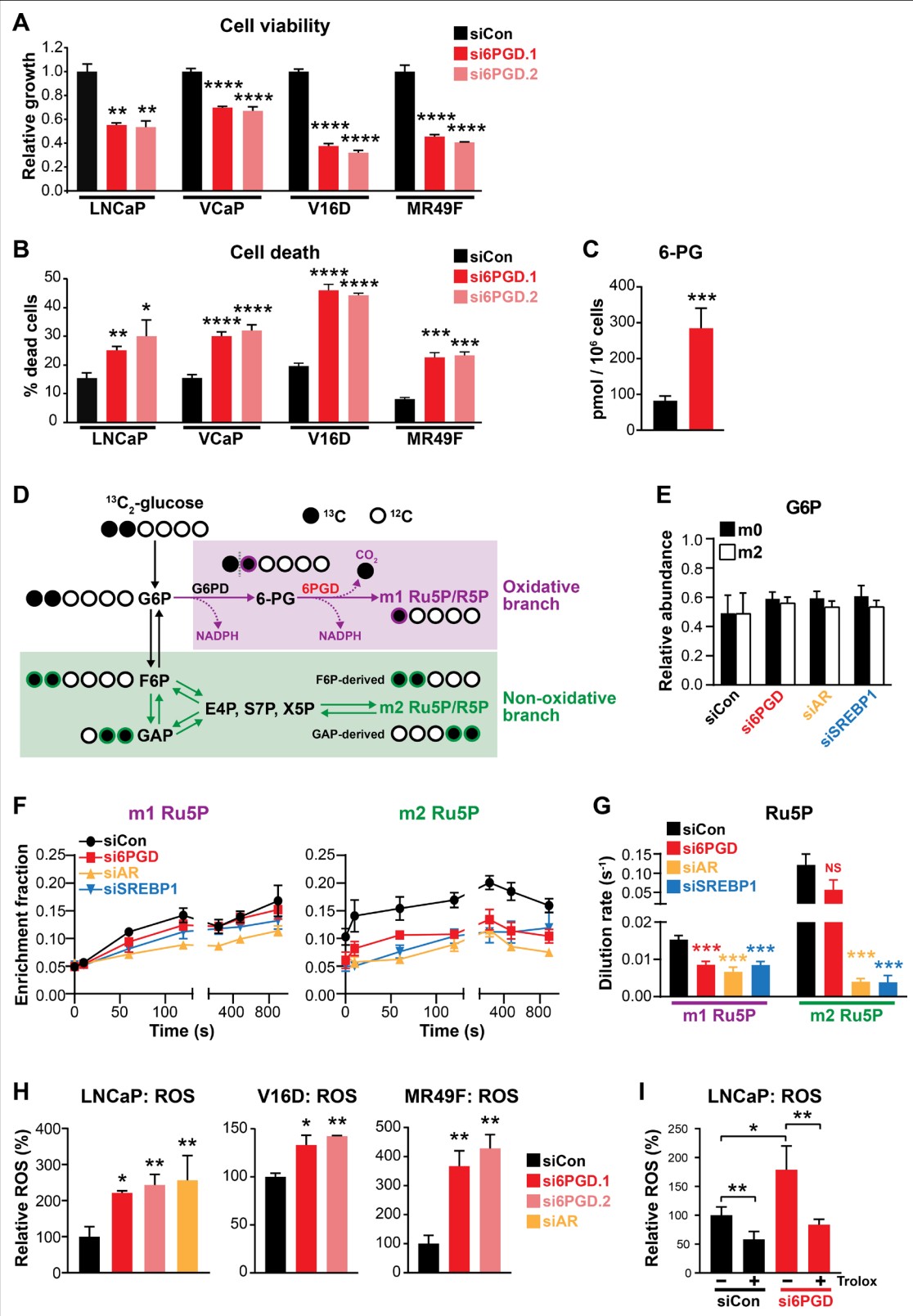

**Figure 3.** An AR-SREBP1-6PGD axis influences prostate cancer cell growth and activity of the pentose phosphate pathway. (**A, B**) Knockdown of 6PGD with two distinct siRNAs (si6PGD.1 and si6PGD.2) reduced viability (**A**) and increased cell death (**B**) of four prostate cancer cell lines, as assessed using Trypan blue exclusion assays. LNCaP and VCaP cells were evaluated 3 days post-transfection; V16D and MR49F cells were evaluated 5 days post-transfection. Error bars are standard error of the mean (SEM) of triplicate samples and are representative of three independent experiments. Treatment

*Figure 3 continued on next page*

*Figure 3 continued*

effects were evaluated using ANOVA and Dunnett's multiple comparison tests (*p<0.05; **p<0.01; ***p<0.001; ****p<0.0001). (**C**) Knockdown of 6PGD causes accumulation of intracellular 6 PG in LNCaP cells, as determined by mass spectrometry. Results are representative of two independent experiments. Error bars are SEM of triplicate samples. Treatment effect was evaluated using an unpaired t test (p<0.001). Colour key is as in (**A**). (**D**) Schematic demonstrating flux of 1,2-$^{13}$C$_2$ glucose through the PPP and incorporation into Ru5P and R5P. Unlabelled $^{12}$C carbon is shown as open circles, whereas $^{13}$C is shown as filled circles. The oxidative and non-oxidative branches of the PPP are indicated in purple and green, respectively. 6 PG: 6-phosphogluconate; E4P: erythrose 4-phosphate; F6P: fructose 6-phosphate; G6P: glucose 6-phosphate; GAP: glyceraldehyde 3-phosphate; R5P: ribose 5-phosphate; Ru5P: ribulose 5-phosphate; S7P: sedoheptulose 7-phosphate; X5P: xylulose 5-phosphate. (**E**) Isotopic steady-state G6P enrichments of LNCaP cells fed with 1,2-$^{13}$C$_2$ glucose and natural glucose at 1:1 ratio show control and treatments cells were labelled to a similar extent. Error bars are standard deviation (SD). (**F**) Accumulation of singly (left, m1) and doubly (right, m2) labelled Ru5P produced via the oxidative and non-oxidative branches, respectively, of the PPP. Error bars are SD. (**G**) Dilution rate (turnover rate) calculated from the accumulation of singly and doubly labelled Ru5P (data from **E**) using the continuous stirred-tank reactor (CSTR) equation. For statistical analysis of treatment effects, refer to Materials and methods (***p<0.001; NS, not significant). Error bars are SD. (**H**) Knockdown of 6PGD and androgen receptor (AR) causes increased levels of reactive oxygen species (ROS) in LNCaP, V16D, and MR49F cells. Data was normalised to siCon, which was set to 100 %. Error bars are SEM of triplicate samples. Treatment effects were evaluated using ANOVA and Dunnett's multiple comparison tests (*p<0.05; **p<0.01). (**I**) ROS production in LNCaP cells in response to si6PGD is reversed by the antioxidant. Trolox data was normalised to siCon in the absence of Trolox, which was set to 100% . Error bars are SEM of triplicate samples. Treatment effects were evaluated using ANOVA and Tukey's multiple comparison tests (*p<0.05; **p<0.01). Colour key is as in (**A**).

The online version of this article includes the following figure supplement(s) for figure 3:

**Figure supplement 1.** Two distinct 6PGD siRNAs (si6PGD.1 and si6PGD.2) effectively reduce 6PGD expression in LNCaP cells.

VCaP and models of CRPC (V16D and MR49F; *Figure 4E,F*). The efficacy of S3 in MR49F cells was particularly notable since this aggressive LNCaP-derived line is resistant to the second-generation AR antagonist Enz (*Kuruma et al., 2013*). S3 was also growth inhibitory in AR-negative PC3 cells, although this line was less sensitive than AR-driven models (*Figure 4—figure supplement 1C*). To assess the potential of targeting 6PGD with S3 in a more clinically relevant setting, we exploited the PDE model (*Centenera et al., 2018*). Notably, S3 reduced proliferation, as measured by IHC for Ki67, in all tumours (n = 9) that were evaluated (*Figure 4G*).

In addition to directly promoting cell growth and survival via anabolism and limiting oxidative stress, the PPP has been reported to suppress AMPK activity by inhibiting its phosphorylation (*Gao et al., 2019*), thereby activating key anabolic pathways mediated by acetyl-CoA carboxylase 1 (ACC1) and mammalian target of rapamycin complex 1 (mTORC1) (*Figure 5A*). Accordingly, we examined whether these pathways are altered in PCa cells in response to 6PGD inhibition. S3 treatment activated AMPK and repressed ACC1 and mTOR pathways in a dose-dependent manner in multiple PCa cell lines, as revealed by increased levels of phospho-AMPK (pAMPK) and phospho-ACC1 (pACC1) and decreased levels of phospho-S6K (pS6K)/phospho-S6 (pS6) (*Figure 5B,C*). Knockdown of 6PGD also repressed ACC1 and mTOR signalling (*Figure 5—figure supplement 1*), verifying that the effects we observed with the inhibitor were on target. More importantly, we recapitulated the impact of S3 on mTOR signalling in our tumour PDE system (*Figure 5C*). Collectively, these results reveal that PPP is an upstream regulator of AMPK, ACC1, and mTOR in PCa; therefore, targeting 6PGD could impede multiple cancer-promoting metabolic pathways.

## A feedback loop between AR and 6PGD supports combinatorial targeting of these factors

During our investigations into the mode of action of S3 and physcion, we noted that both agents reduced steady-state levels of AR protein in models of castration-sensitive and castration-resistant PCa (*Figure 6A*, *Figure 6—figure supplement 1A*). This observation suggested that targeting 6PGD would inhibit the AR signalling axis. We validated this hypothesis by demonstrating that S3 and physcion dose-dependently reduced the expression of AR and its target genes in multiple cell line models (*Figure 6A,B*, *Figure 6—figure supplement 1B–D*) and, importantly, in our clinical PDE tissues (*Figure 6C*). Although 6PGD inhibitors significantly decreased AR protein, they did not alter *AR* transcript levels (*Figure 6B*, *Figure 6—figure supplement 1B–D*), indicative of a post-transcriptional mechanism. Since the ubiquitin-proteasome system (UPS) plays an integral role in AR protein stability (*Wen et al., 2020*), we hypothesised that 6PGD inhibition could enhance AR ubiquitylation and turnover. To test this idea, LNCaP cells were treated with a combination of S3 and the proteasome inhibitor MG132, after which the levels of total and ubiquitylated AR were measured by western blotting.

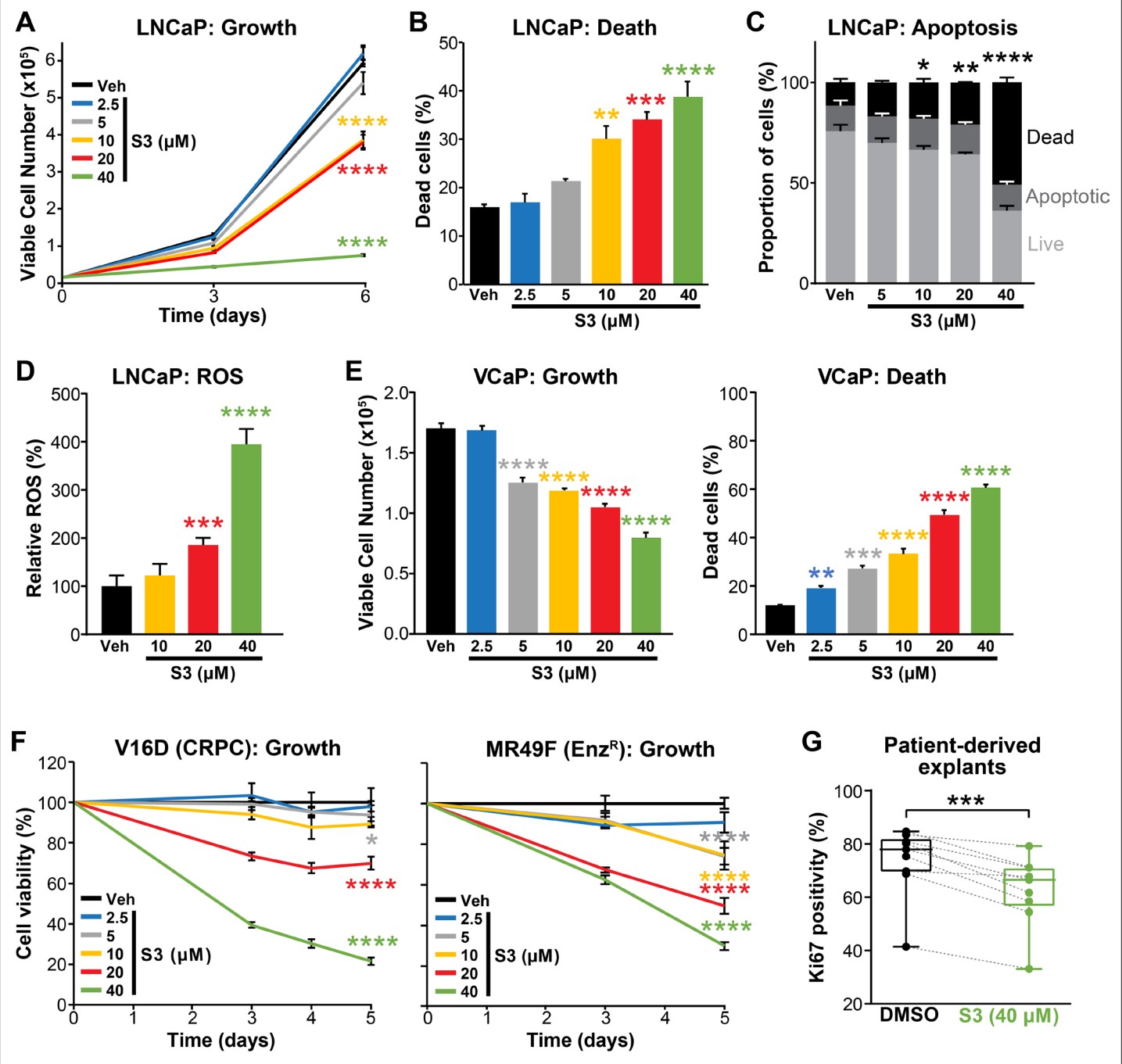

**Figure 4.** Pharmacological inhibition of 6PGD suppresses prostate cancer growth and increases reactive oxygen species (ROS). (**A, B**) The 6PGD inhibitor, S3, dose-dependently decreased viability (**A**) and increased death (**B**) of LNCaP cells, as determined by Trypan blue exclusion assays. Dead cells were counted at day 6. Data represent the mean of triplicate samples and are representative of three independent experiments. Error bars are SEM. Growth (day 6) and death for each dose was compared to vehicle using ANOVA and Dunnett's multiple comparison tests (****$p<0.0001$). Veh: vehicle. (**C**) S3 causes apoptosis of LNCaP cells, as determined using flow cytometry-based Annexin V/7-AAD assays. Cells were assessed 72 hr after treatment. Data represent the mean ± SE of triplicate samples and are representative of four independent experiments. Dead cell proportions were compared to vehicle using ANOVA and Dunnett's multiple comparison tests (*$p<0.05$; **$p<0.01$; ****$p<0.0001$). (**D**) S3 causes increased levels of ROS in LNCaP cells. Data was normalised to Veh, which was set to 100% . Effects were evaluated using ANOVA and Dunnett's multiple comparison tests (***$p<0.001$; ****$p<0.0001$). (**E**) S3 dose-dependently decreased viability (left) and increased death (right) of VCaP cells, as determined by Trypan blue exclusion assays. Live and dead cells were counted 4 days after treatment. Data represent the mean ± SE of triplicate samples and are representative of three independent experiments. Effects were evaluated using ANOVA and Dunnett's multiple comparison tests (**$p<0.01$; ***$p<0.001$; ****$p<0.0001$). (**F**) S3 suppresses the growth of castration-resistant prostate cancer (CRPC) cells (V16D) and enzalutamide-resistant CRPC cells (MR49F), as determined

*Figure 4 continued on next page*

*Figure 4 continued*

using CyQuant Direct Cell Proliferation Assay. Fluorescence from day 0 was set to 100% . Data represent the mean ± SEM of triplicate samples and are representative of two independent experiments. Effects (at day 5) were evaluated using ANOVA and Dunnett's multiple comparison tests (*p<0.05; ****p<0.0001). (**G**) S3 inhibits the proliferation of prospectively collected human tumours grown as patient-derived explants (PDEs). PDEs (from n = 9 patients) were treated for 72 hr. Ki67 positivity, a marker of proliferation, was determined using immunohistochemistry. Boxes show minimum and maximum (bottom and top lines, respectively) and mean (line within the boxes) values. A paired t test was used to compare Ki67 positivity in treated versus vehicle-treated control samples (***p<0.001).

The online version of this article includes the following figure supplement(s) for figure 4:

**Figure supplement 1.** Physcion effectively suppresses growth (**A**) and causes death (**B**) of LNCaP cells.

In the presence of MG132, accumulation of ubiquitylated AR as well as the total cellular ubiquitylated protein pool was evident in S3-treated cells (***Figure 6D***). Moreover, in the presence of MG132, S3 did not reduce total AR protein levels beyond that caused by MG132 alone (***Figure 6D***). Collectively, these findings indicate that inhibition of 6PGD by S3 enhances turnover of AR by the UPS.

Our results demonstrated that AR induces *6PGD* gene expression (via SREBP1) and that 6PGD can enhance the stability of AR protein, collectively revealing a positive feedback loop between androgen signalling and the PPP. The co-dependency of these pathways led us to speculate that a combinatorial targeting approach could be an effective PCa therapy. In support of this hypothesis, enzalutamide and S3 exhibited an additive effect in androgen-sensitive (VCaP) and CRPC (V16D) cell lines (***Figure 6E–G***). Collectively, these findings highlight the complex interplay between AR and 6PGD in PCa cells and identify a potential new combinatorial therapy.

## Discussion

PCa possesses a unique androgen-regulated metabolic profile, characterised by high rates of lipogenesis and oxidative phosphorylation compared to the normal state. More recently, altered glucose metabolism has emerged as another feature of this common malignancy (***Lin et al., 2019***). In this study, we identified *6PGD* as an AR-regulated gene that may not be effectively suppressed in tumour cells by current ARPIs such as Enz. 6PGD is the third enzyme in a critically important glucose metabolic pathway, the PPP. Our data reveal that a positive feedback loop between AR and 6PGD enhances growth and survival of tumour cells. This work not only expands our knowledge of the interplay between hormones and glucose metabolism in PCa but also exposes a new therapeutic vulnerability.

Our identification of 6PGD as an androgen-regulated PPP enzyme lends further support to this pathway being a key metabolic target of androgens in PCa. Frigo and colleagues recently demonstrated that G6PD, the rate-limiting enzyme of this pathway, is also transcriptionally and post-transcriptionally regulated by AR signalling (***Tsouko et al., 2014***). Moreover, an enzyme that regulates the non-oxidative phase of the PPP, transketolase-like protein 1 (TKTL1), increases in expression during PCa progression, being highest in metastatic tumours (***da Costa et al., 2018***). Such multi-level control of a single pathway emphasises the relevance of increased PPP flux in PCa. It is notable that the androgen-regulated enzymes of this pathway, 6PGD and G6PD, both catalyse steps in the NADPH-generating oxidative phase of the PPP; this represents another mechanism underlying hormonal protection against oxidative stress in the prostate.

Despite its role as a key downstream effector of androgen-regulated cellular metabolism, our data do not support a direct mode of transcriptional regulation of *6PGD* by AR. Rather, AR harnesses another key metabolic transcription factor, SREBP1, to drive expression of 6PGD and hence activity of the PPP. SREBP1, a transcription factor that regulates genes involved in fatty acid and cholesterol biosynthesis and homeostasis that is activated and upregulated by AR signalling (***Heemers et al., 2006***; ***Duvel et al., 2010***), is itself a therapeutic target in PCa (***Galbraith et al., 2018***). Some metabolic genes appear to be directly co-regulated by AR and SREBP1 based on the binding of both factors to cis-regulatory elements (e.g. *FASN*, ***Chan et al., 2015***; ***Choi et al., 2008***). However, our observation that siRNA or pharmacological targeting of SREBP1 blocks androgen-mediated induction of *6PGD* suggests that SREBP1 transcriptionally activates this gene downstream of AR. Further supporting the relevance of a closely interlinked AR/SREBP1/6PGD pathway in PCa was our observation that targeting any one of these three factors had a pronounced impact on glucose flux through the oxidative branch of the PPP. Interestingly, knockdown of AR and SREBP1 also had a profound impact on the

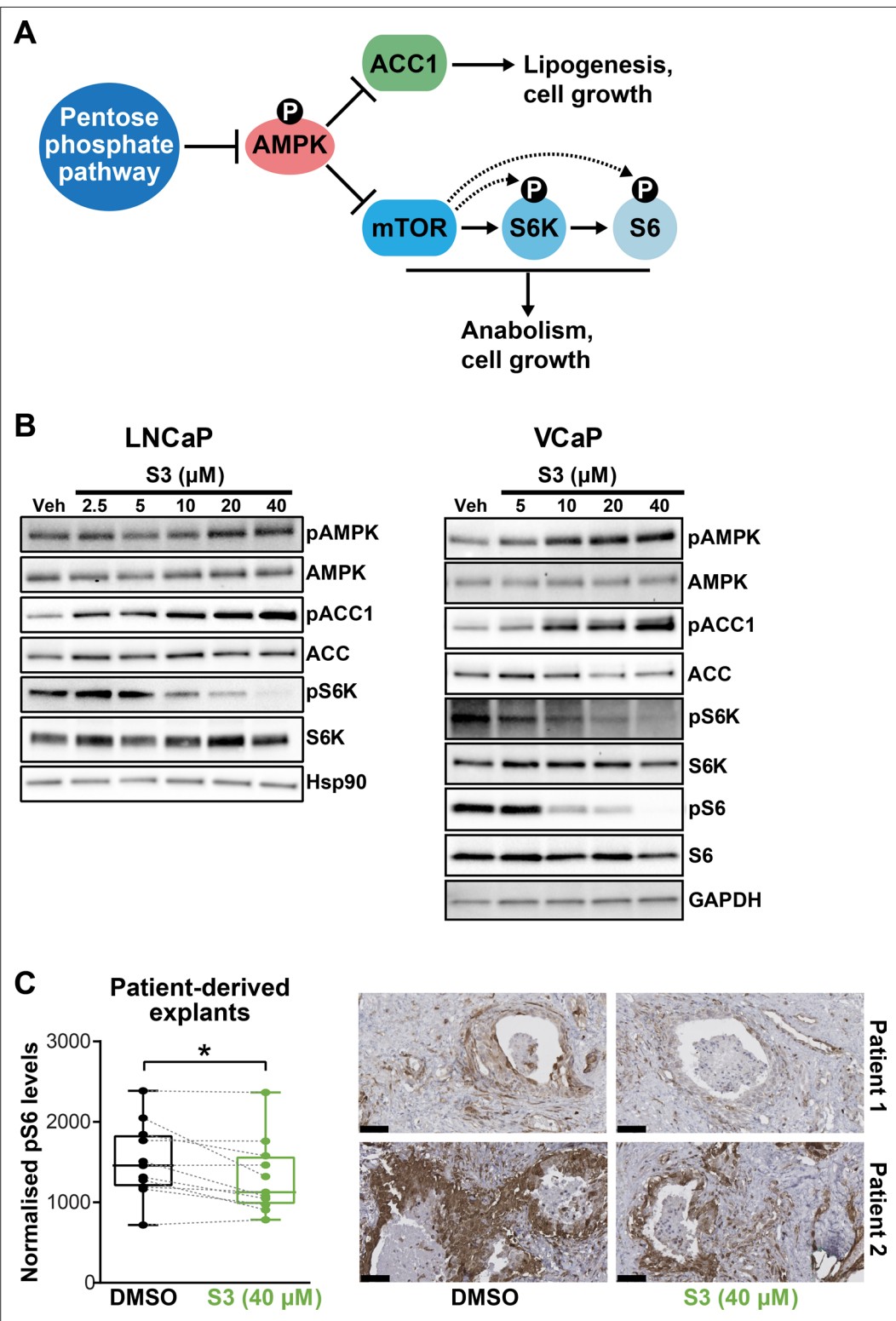

**Figure 5.** Targeting 6PGD activates AMPK and represses ACC1 and mTOR pathways. (**A**) Schematic showing key metabolic pathways downstream of the pentose phosphate pathway (PPP). By suppressing AMPK signalling, the PPP can enhance the activity of ACC1 and mTOR and subsequently various growth-promoting anabolic processes. (**B**) S3 activates AMPK and inhibits ACC1 and mTOR signalling. LNCaP (left) and VCaP (right) cells were treated for 24 hr with the indicated doses of S3 prior to analysis of indicated proteins by immunoblotting. (**C**) S3 inhibits mTOR signalling, as indicated by reduced pS6, in patient-derived explants (PDEs). PDEs (from n = 11 patients)

*Figure 5 continued on next page*

*Figure 5 continued*

were treated for 72 hr. The levels of pS6 were measured using immunohistochemistry (IHC). Boxes (graph on left) show minimum and maximum (bottom and top lines, respectively) and mean (line within the boxes) values. A paired t test was used to compare Ki67 positivity in treated versus vehicle-treated control samples (***p<0.001). Representative IHC images are shown on the right (scale bars represent 50 μm).

The online version of this article includes the following figure supplement(s) for figure 5:

**Figure supplement 1.** 6PGD knockdown inhibits ACC1 and mTOR signalling, as determined by increased levels of pACC1 and decreased levels of pS6/pS6K, respectively.

non-oxidative branch of the PPP. Although the precise mechanism(s) underlying this observation are not known, given their expansive and diverse roles in PCa cell metabolism, it is plausible that AR and SREBP1 regulate other metabolic factors that stimulate the non-oxidative PPP. Importantly, regulation of the PPP by AR-SREBP1-6PGD has a broader clinical implication; therapeutic strategies that effectively suppress this pathway would impinge on the activity of three important oncogenic drivers with multifaceted cancer-promoting activities.

We propose that AR-mediated activation of the PPP in PCa would yield additional advantages beyond the generation of key substrates for nucleic acid anabolism and the antioxidant NADPH. Most notably, PPP suppression of AMPK, itself a hub for cellular metabolic and growth control, results in augmentation of ACC1 and mTOR activity (*Zadra et al., 2014*). The importance of both ACC1 and mTOR in enabling PCa cells to meet their energy demands is increasingly well recognised; indeed, both of these factors are key mediators of de novo lipogenesis, high levels of which are a hallmark of prostate tumours (*Mah et al., 2019*). Mechanistically, it has been reported that 6PGD-mediated production of Ru5P inhibits AMPK by disrupting the LKB1 complex, leading to activation of ACC1 and lipogenesis (*Lin et al., 2015*). Thus, in addition to its more direct impact on lipogenesis by regulation of lipid metabolic genes (*Mah et al., 2019*), our data reveal that AR also supports this metabolic process by activation of 6PGD and the PPP.

In addition to regulation of 6PGD by the androgen signalling axis, our work also revealed that 6PGD can act in a reciprocal manner to maintain AR protein levels and activity. Indeed, S3 was as effective as Enz at inhibiting the expression of some AR target genes, albeit at higher doses. We propose that this positive feedback would serve as an effective circuit to fuel PCa growth and enhance survival. Mechanistically, we demonstrated that targeting of 6PGD results in increased ubiquitylation of AR, explaining why it is decreased at the protein level. Precisely how 6PGD inhibition regulates processing of AR by the UPS is unclear. However, we note that S3 treatment increased ROS levels and activated AMPK signalling, both of which have been shown to promote AR degradation/turnover (*Wu et al., 2018*; *Shen et al., 2014*). Thus, we propose that 6PGD regulation of AR protein ubiquitylation, and hence stability, likely occurs at multiple levels. More broadly, unravelling the complexity of the AR/6PGD feedback loop will be important to effectively harness co-targeting strategies.

Given the important role of the PPP in PCa growth and survival, established by this study in addition to earlier work (*Tsouko et al., 2014*; *Ros et al., 2012*), targeting this pathway as a possible therapeutic strategy has merit. We investigated this concept using two inhibitors of 6PGD, physcion (1,8-dihydroxy-3-methoxy-6-methyl-anthraquinone; emodin-3-methyl ether) and S3 (1-hydroxy-8-methoxy-anthraquinone). Physcion (also known as parietin; PubChem CID 10639) was the most active inhibitor of 6PGD activity in an in vitro assay amongst a library of ~2000 small molecules (*Lin et al., 2015*). A plant-derived anthraquinone, physcion was initially investigated for its anti-microbial and anti-inflammatory activities (*XunLi et al., 2019*). More recently, there has been significant interest in its repurposing as an oncology agent since it has been reported to possess broad anti-cancer activity (i.e. suppression of growth and migration, induction of apoptosis) in leukaemia, colorectal, cervical, and breast cancer cells, amongst others (*Lin et al., 2015*; *Hong et al., 2014*; *Chen et al., 2015*; *Elf et al., 2017*; *Pan et al., 2018*). However, while physcion has achieved impressive anti-cancer results in some preclinical studies, its poor pharmacological attributes, including low solubility, may impede efforts to progress it to the clinic (*XunLi et al., 2019*). Therefore, we also tested the physcion derivative compound S3, which has been reported to possess improved pharmacological attributes (*Lin et al., 2015*). Our results represent the first evaluation of physcion and S3 in PCa and collectively highlight the potential of therapeutically targeting 6PGD in this disease. Indeed, our data suggest that S3/physcion would possess multi-pronged anti-tumour activity in PCa by inhibiting oncogenic

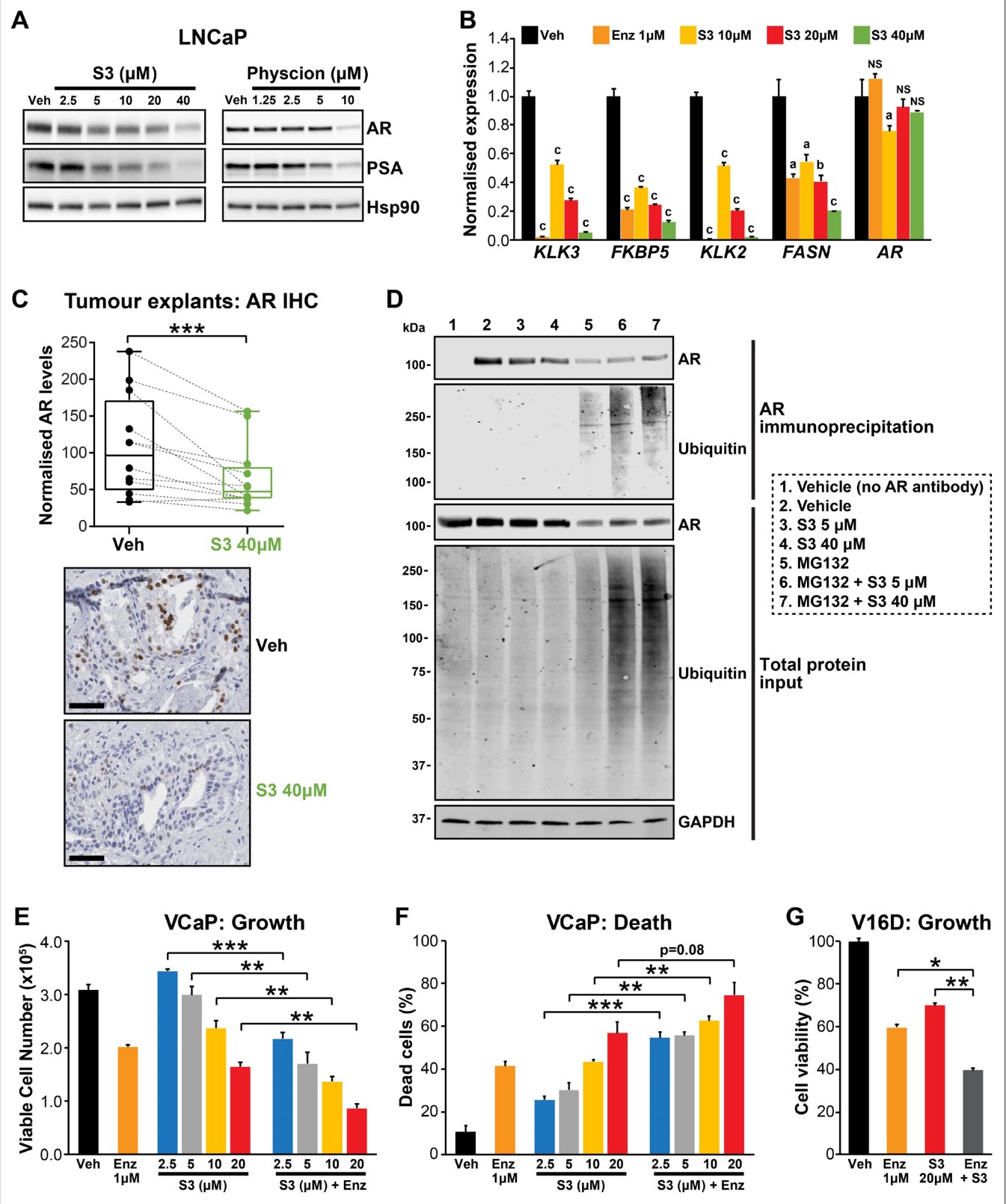

**Figure 6.** Targeting the androgen receptor (AR)/PGD feedback loop in prostate cancer. (**A**) Protein levels of AR and its target PSA in response to S3 (24 hr of treatment) and physcion (48 hr of treatment) in LNCaP cells, as determined by immunoblotting. HSP90 was used as a loading control. (**B**) AR target gene expression in response to S3 treatment in LNCaP cells, as determined by RT-qPCR. Gene expression was normalised to *GUSB* and *L19* and represents the mean + SEM of three biological replicates; Veh was set to 1. Differential expression was evaluated using ANOVA and Dunnett's

*Figure 6 continued on next page*

*Figure 6 continued*

multiple comparison tests (a, p<0.01; b, p<0.001; c, p<0.0001; NS, not significant). (**C**) S3 reduces AR protein levels in patient-derived explants (PDEs). AR levels in tumours from 14 patients were measured by immunohistochemistry (IHC; left). Boxes show minimum and maximum (bottom and top lines, respectively) and mean (line within the boxes) values. A paired t test was used to compare AR levels in treated versus control samples (***p<0.001). Representative IHC images are shown on the right (scale bars represent 50 µm). (**D**) S3 enhances AR ubiquitylation. LNCaP cells were treated with indicated concentrations of S3 ±10 µM MG132, or 10 µM MG132 alone, for 24 hr prior to AR immunoprecipitation. Both immunoprecipitates and total protein inputs (1/30 of immunoprecipitates) were subjected to immunoblotting analysis for the indicated proteins. (**E, F**) Anti-cancer effects of combined Enz and S3 treatment in VCaP cells. Live (**E**) and dead (**F**) cells were measured by Trypan blue exclusion assays 4 days after treatment. Data represent the mean + SEM of triplicate samples and are representative of three independent experiments. (**G**) Anti-cancer effects of combined Enz and S3 treatment in V16D cells. Live cells (**F**) were measured as in (**D**) after 3 days of treatment; data are representative of three independent experiments.

The online version of this article includes the following figure supplement(s) for figure 6:

**Figure supplement 1.** 6PGD inhibitors suppress AR expression and activity.

metabolism, including lipogenesis (i.e. activation of AMPK and suppression of ACC1 and mTOR); increasing levels of ROS, resulting in oxidative stress and lipid peroxidation; and finally, suppressing the levels and activity of AR, the primary oncogenic driver of this disease. Importantly, a Phase I trial reported that physcion was well tolerated with low toxicity (*Tzeng et al., 2011*), supporting its future clinical application.

Since AR-targeted therapies are not curative, there is intense interest in identifying combination therapies that would improve patient outcomes. Our work provides a solid rationale for co-targeting of AR and 6PGD; indeed, we observed synergistic effects of Enz and S3 in PCa models. Moreover, the existence of an AR:6PGD feedback loop enhances the appeal of such a combinatorial strategy. Although we acknowledge that physcion and S3 may not be useful clinical agents due to pharmacological issues, we expect that the future development of therapies that effectively suppress activity of 6PGD, or other components of the PPP, could have a major impact on PCa patients.

## Materials and methods

### Key resources table

| Reagent type (*species*) or resource | Designation | Source or reference | Identifiers | Additional information |
|---|---|---|---|---|
| Cell line (*Homo-sapiens*) | LNCaP | ATCC | ATCC CRL-1740 (RRID:CVCL_1379) | |
| Cell line (*Homo-sapiens*) | VCaP | ATCC | ATCC CRL-2876 (RRID:CVCL_2235) | |
| Cell line (*Homo-sapiens*) | PC3 | ATCC | ATCC CRL-7934 (RRID:CVCL_0035) | |
| Cell line (*Homo-sapiens*) | 22RV1 | ATCC | ATCC CRL-2505 (RRID:CVCL_1045) | |
| Cell line (*Homo-sapiens*) | V16D | PMID:27046225 | Kind gift from Prof. Amina Zoubeidi | |
| Cell line (*Homo-sapiens*) | MR49F | PMID:27046225 | Kind gift from Prof. Amina Zoubeidi | |
| Transfected construct (*Homo sapiens*) | Negative control siRNA | Ambion; Thermo Fisher Scientific | AM4637 | |
| Transfected construct (*Homo sapiens*) | siAR | Thermo Fisher Scientific | Silencer Select 4390824 | |
| Transfected construct (*Homo sapiens*) | siAR | Thermo Fisher Scientific | Silencer Select 4390825 | |
| Transfected construct (*Homo sapiens*) | siAR | Thermo Fisher Scientific | Custom 4399665 | |
| Transfected construct (*Homo sapiens*) | si6PGD | Thermo Fisher Scientific | 4427038 | |
| Tansfected construct (*Homo sapiens*) | siSREBP1 | Dharmacon | ON-TARGETplus 6720 | |

*Continued on next page*

*Continued*

| Reagent type (*species*) or resource | Designation | Source or reference | Identifiers | Additional information |
|---|---|---|---|---|
| Antibody | ACC-1 [C83B10] (rabbit monoclonal) | Cell Signaling Technology, Inc | 3676 (RRID:AB_2219397) | Western blot: (1:1000) |
| Antibody | pACC-1 [Ser79] (rabbit polyclonal) | Cell Signaling Technology, Inc | 3661 (RRID:AB_330337) | Western blot: (1:1000) |
| Antibody | β-Actin (AC-15) (mouse monoclonal) | Sigma Aldrich | A5441 (RRID:AB_476744) | Western blot: (1:1000) |
| Antibody | AR-N20 (rabbit polyclonal) | Santa Cruz Biotechnology Inc | sc-816 (RRID:AB_1563391) | Western blot: (1:1000) |
| Antibody | AR (rabbit monoclonal) | Abcam | ab108341 (RRID:AB_10865716) | Immunohistochemistry: (1:200) |
| Antibody | AR (mouse monoclonal) | Santa Cruz Biotechnology Inc | sc-7305 (RRID:AB_626671) | Immunoprecipitation: (0.2 µg) |
| Antibody | AMPKα (rabbit polyclonal) | Cell Signaling Technology, Inc | 2532 (RRID:AB_330331) | Western blot: (1:1000) |
| Antibody | pAMPKα [Thr172] 40H9 (rabbit monoclonal) | Cell Signaling Technology, Inc | 2535 (RRID:AB_331250) | Western blot: (1:1000) |
| Antibody | GAPDH (HuCAL recombinant) | BioRad | 12004168 | Western blot: (1:1000) |
| Antibody | Hsp90 (rabbit polyclonal) | Cell Signaling Technology, Inc | 4874 (RRID:AB_2121214) | Western blot: (1:1000) |
| Antibody | Ki67 (mouse monoclonal) | Agilent Technologies | M724001-2 (RRID:AB_2631211) | Immunohistochemistry: (1:200) |
| Antibody | P70 S6 Kinase (49D7) (rabbit monoclonal) | Cell Signaling Technology, Inc | 2708 (RRID:AB_390722) | Western blot: (1:1000) |
| Antibody | pP70 S6 Kinase [Thr389] (rabbit polyclonal) | Cell Signaling Technology, Inc | 9205S (RRID:AB_330944) | Western blot: (1:2000) |
| Antibody | 6PGD (rabbit polyclonal) | ThermoFisher Scientific | PA5-21376 (RRID:AB_11153623) | Western blot: (1:1000) |
| Antibody | 6PGD (rabbit polyclonal) | Sigma Aldrich | HPA031314 (RRID:AB_10610278) | Immunohistochemistry: (1:800) |
| Antibody | PSA (rabbit polyclonal) | ProteinTech Group | 10679-1-AP (RRID:AB_2134244) | Western blot: (1:1000) |
| Antibody | S6 (5G10) (rabbit monoclonal) | Cell Signaling Technology, Inc | 2217 (RRID:AB_331355) | Western blot: (1:1000) |
| Antibody | pS6 [Ser235/236] (rabbit polyclonal) | Cell Signaling Technology, Inc | 2211 (RRID:AB_331679) | Western blot: (1:1000) Immunohistochemistry: (1:200) |
| Antibody | Ubiquitin (mouse monoclonal) | Genesearch | 3936 | Western blot: (1:1000) |
| Antibody | Goat Anti-Rabbit (Biotinylated) | Agilent Technologies | E043201-8 | Immunohistochemistry: (1:400) |
| Sequence-based reagent | AR For | This paper | qRT-PCR primers | CAACTCCTTCAGCAACAGCA |
| Sequenced-based reagent | AR Rev | This paper | qRT-PCR primers | TCGAAGTGCCCCCTAAGTAA |
| Sequence-based reagent | FKBP5 For | This paper | qRT-PCR primers | AAAAGGCCAAGGAGCACAAC |
| Sequenced-based reagent | FKBP5 Rev | This paper | qRT-PCR primers | TTGAGGAGGGGCCGAGTTC |

*Continued on next page*

*Continued*

| Reagent type (*species*) or resource | Designation | Source or reference | Identifiers | Additional information |
|---|---|---|---|---|
| sSquence-based reagent | GAPDH For | This paper | qRT-PCR primers | TGCACCACCAACTGCTTAGC |
| Sequenced-based reagent | GAPDH Rev | This paper | qRT-PCR primers | GGCATGGACTGTGGTCATGAG |
| Sequence-based reagent | GUSB For | This paper | qRT-PCR primers | CGTCCCACCTAGAATCTGCT |
| sSquenced-based reagent | GUSB Rev | This paper | qRT-PCR primers | TTGCTCACAAAGGTCACAGG |
| Sequence-based reagent | KLK2 For | This paper | qRT-PCR primers | GGTGGCTGTGTACAGTCATGGAT |
| Sequenced-based reagent | KLK2 Rev | This paper | qRT-PCR primers | TGTCTTCAGGCTCAAACAGGTTG |
| Sequence-based reagent | KLK3 For | This paper | qRT-PCR primers | ACCAGAGGAGTTCTTGACCCCAAA |
| Sequenced-based reagent | KLK3 Rev | This paper | qRT-PCR primers | CCCCAGAATCACCCGAGCAG |
| Sequence-based reagent | L19 For | This paper | qRT-PCR primers | TGCCAGTGGAAAAATCAGCCA |
| Sequenced-based reagent | L19 Rev | This paper | qRT-PCR primers | CAAAGCAAATCTCGACACCTTG |
| Sequenced-based reagent | PGD For | This paper | qRT-PCR primers | CACAGCAGGGTTCTCCAGTT |
| Sequenced-based reagent | PGD Rev | This paper | qRT-PCR primers | GTCAGTGGTGGAGAGGAAGG |
| Sequenced-based reagent | PPIA For | This paper | qRT-PCR primers | GCATACGGGTCCTGGCAT |
| Sequenced-based reagent | PPIA Rev | This paper | qRT-PCR primers | ACATGCTTGCCATCCAACC |
| Sequenced-based reagent | TMPRSS2 For | This paper | qRT-PCR primers | GACCAAGAACAATGACATTGCG |
| Sequenced-based reagent | TMPRSS2 Rev | This paper | qRT-PCR primers | GTTCTGGCTGCAGCATCATG |
| Sequenced-based reagent | TUBA1B For | This paper | qRT-PCR primers | CCTTCGCCTCCTAATCCCTA |
| Sequenced-based reagent | TUBA1B Rev | This paper | qRT-PCR primers | CCGTGTTCCAGGCAGTAGA |
| Chemical compound, drug | Dihydrotestosterone | Sigma Aldrich | Cas#: 521-18-6 | |
| Chemical compound, drug | Enzalutamide | Selleck Chemicals | Cat#: S1250 | |
| Chemical compound, drug | Apalutamide | Selleck Chemicals | Cat#: S2840 | |
| Chemical compound, drug | Darolutamide | Selleck Chemicals | Cat#: S7559 | |
| Chemical compound, drug | S3 | Sigma Aldrich | Cat#: R164046 | |
| Chemical compound, drug | Physcion | Sigma Aldrich | Cat#: 93893 | |

*Continued on next page*

*Continued*

| Reagent type (*species*) or resource | Designation | Source or reference | Identifiers | Additional information |
|---|---|---|---|---|
| Chemical compound, drug | Trolox | Selleck Chemicals | Cat#: S3665 | |
| Chemical compound, drug | 1,2-13C2 glucose | Sigma Aldrich | Cat#: 453188 | |
| Commercial assay or kit | RNeasy Mini extraction kit | Qiagen | Cat#: 74104 | |
| Commercial assay or kit | iScriptTM cDNA Synthesis kit | Bio-Rad | Cat#: 1708890 | |
| Commercial assay or kit | NEXTflex Rapid Illumina Directional RNA-Seq Library Prep Kits | Perkin-Elmer | Cat#: NOVA-5138 | |
| Commercial assay or kit | CyQuant Assay Cell Proliferation Assays | Thermo Fisher Scientific | Cat#: C7026 | |
| Commercial assay or kit | CellROX Orange Flow Cytometry Assay Kits | Life Technologies | Cat#: C10493 | |
| Other | Lipofectamine RNAiMAX transfection reagent | Thermo Fisher Scientific | 13778075 | |
| Software, algorithm | GraphPad Prism | GraphPad Software, Inc. | Prism V7 RRID:SCR_002798 | |
| Software, algorithm | R | R Core Team (2019) | R version 3.6.2 RRID:SCR_001905 | |
| Software, algorithm | ImageJ analysis software | NIH | ImageJ RRID:SCR_003070 | |
| Software, algorithm | TraceFinder v5.0 | Thermo Fisher Scientific | OPTON-30688 | |

## Reagents

Chemicals, solvents, and solutions, including physcion ($C_{16}H_{12}O_5$; 1,8-dihydroxy-3-methoxy-6-methyl-anthraquinone; emodin-3-methyl ether) and S3 ($C_{15}H_{10}O_4$; 1-hydroxy-8-methoxy-anthraquinone), were obtained from Sigma-Aldrich (St Louis, MO, USA), except for enzalutamide (Selleck Chemicals; Houston, TX, USA); apalutamide (ARN-509), darolutamide (ODM-201), and Trolox (Sapphire Bioscience; Redfern, NSW, Australia). All chemicals/reagents were dissolved in dimethyl sulfoxide (DMSO) except dihydrotestosterone (DHT), which was dissolved in ethanol.

## Cell line models

LNCaP (RRID:CVCL_1379), VCaP (RRID:CVCL_2235), PC3 (RRID:CVCL_0035), and 22Rv1 (RRID:CVCL_1045) human prostate carcinoma cells were obtained from the American Type Culture Collection (ATCC, MD, USA). Dr. Amina Zoubeidi (Vancouver Prostate Centre, Vancouver, Canada) kindly provided LNCaP-V16D (castration-resistant, enzalutamide-sensitive) and LNCaP-MR49F (castration-resistant, enzalutamide-resistant) human PCa cells (*Kuruma et al., 2013*). LNCaP, 22Rv1, V16D, and MR49F cells were maintained in RPMI-1640 containing 10% FBS; the media for growth of MR49F cells was additionally supplemented with 10 µM Enz. VCaP cells were maintained in Dulbecco's Modified Eagle's Medium containing 10% FBS, 1% sodium pyruvate, 1% MEM non-essential amino acids, and 0.1 nM 5α-dihydrotestosterone (DHT). PC3 cells were maintained in RPMI-1640 containing 5% FBS. All cell lines were authenticated using short tandem repeat profiling in 2018/2019 by ATCC or CellBank Australia, and undergo regular testing for mycoplasma contamination.

## Transfection of PCa cell lines

Gene-specific knockdown was achieved by reverse-transfection of PCa cell suspensions (total $5 \times 10^5$ cells) with 12.5 nM siRNA in six-well plates using RNAiMAX transfection reagent (Life Technologies;

Thermo Fisher Scientific, Scornsby, VIC, Australia), according to the manufacturer's instructions. The siRNAs used in this study were AR (Silencer Select #4390824/5; s1538, s1539 and custom #4399665; s551824 (sense: GAACUUCGAAUGAACUACATt, antisense: UGUAGUUCAUUCGAAGUUCat)), 6PGD (Silencer Select #4427038; s10394 and 10395; Thermo Fisher Scientific), SREBP1 (ON-TARGETplus 6720; Dharmacon), and Negative Control 2 #AM4637 (Ambion; Thermo Fisher Scientific).

## Quantitative real-time PCR

Reverse transcription of (1 μg) and qPCR was done as described previously (*Gillis et al., 2013*). GeNorm (*Vandesompele et al., 2002*) was used to identify suitable reference genes: gene expression in cell lines is presented relative to *L19* and *GUSB*, and gene expression in prostate tumour explants is presented relative to *GAPDH*, *PPIA* and *TUBAIB*. Primer sequences are provided in *Supplementary file 1*.

## Immunoblotting

Whole-cell lysates were prepared using RIPA buffer containing cOmplete ULTRA protease and phosphatase inhibitor (Cell Signaling Technology [CST], Danvers, MA, USA) and Western blotting was performed as described previously (*Armstrong et al., 2018*). A list of primary and secondary antibodies used in the study is provided in *Supplementary file 2*.

## RNA sequencing (RNA-seq)

LNCaP cells were seeded at density $5 \times 10^5$ cells in six-well dishes (Corning) and treated with 1 μM Enz (or 0.1% DMSO control) or transfected with 12.5 nM AR siRNA (or scrambled siRNA control). Each treatment comprised four replicates. After 24 hr, the cells were collected in Trizol (four replicates, for RNA analysis) or RIPA buffer + protease inhibitors (two replicates, for protein analysis). RNA extractions were completed using RNeasy Mini spin columns (Qiagen, Chadstone, VIC, Australia), according to the manufacturer's instructions. RNA was eluted in 40 μl RNase-free $H_2O$. RT-qPCR and western blotting were performed to verify the expected response of known AR-regulated proteins and genes, PSA/*KLK3* and FKBP51/*FKBP5*. Subsequently, libraries were generated using 800 ng of RNA and NEXTflex Rapid Illumina Directional RNA-Seq Library Prep Kits (Bio-Scientific, Kirrawee, NSW, Australia), according to the manufacturer's instructions. Sequencing was carried out at the South Australian Health and Medical Research Institute Genomics Facility using an Illumina NextSeq 500 (single read 75 bp v2 sequencing chemistry). The quality and number of reads for each sample were assessed with FastQC v0.11.3 (*Andrews, 2010*). Adaptors were trimmed from reads, and low-quality bases, with Phred scores < 28, were trimmed from ends of reads, using Trimgalore v0.4.4 (*Krueger, 2012*). Trimmed reads of <20 nucleotides were discarded. Reads passing all quality control steps were aligned to the hg38 assembly of the human genome using TopHat v2.1.1 (*Kim et al., 2013*) allowing for up to two mismatches. Reads not uniquely aligned to the genome were discarded. HTSeq-count v0.6.1 (*Anders et al., 2015*) was used with the union model to assign uniquely aligned reads to Ensembl Hg38.86-annotated genes. Data were normalised across libraries by the trimmed mean of M-values (TMM) normalisation method, implemented in the R v3.5.0, using Bioconductor v3.6 EdgeR v3.20.9 package (*Robinson et al., 2010*). Only genes expressed at count-per-million value greater than 10 in at least two samples per group were retained for further analysis. Differentially expressed genes were selected based on the robust version of the quasi-likelihood negative binomial generalised log-linear model (*Lun et al., 2016*), with false discovery rate (FDR) set at 0.05. RNA-seq data are available through NCBI's Gene Expression Omnibus (GSE152254).

## Cell growth and apoptosis assays

Cell growth curves were done using Trypan blue exclusion and manual counting of cells, as described previously (*Centenera et al., 2015*). Cell viability was also determined by CyQuant Assay Cell Proliferation Assays (Thermo Fisher Scientific), according to the manufacturer's instructions. Apoptosis was measured by collecting cells in FACS binding buffer (47 ml of HANKS buffered saline, 500 μl of Herpes solution, and 2.5 ml of 100 mM $CaCl_2$), staining with Annexin V PE BD Pharmingen (BD Biosciences, CA, USA) and 1 mM 7-aminoactiomycin D (Thermo Fisher Scientific) and analysis by Flow Cytometry using a BD LSRFortessa X20.

## Metabolomics

To measure 6 PG abundance (*Figure 3C*), LNCaP cells were seeded at a density of $5 \times 10^5$ cells per well into Nunclon D multi-dishes with poly-lysine coating (Thermo Fisher Scientific), with or without transfection with siPGD (Silencer Select s10394). At time of collection, cells were washed twice with 0.9% w/v NaCl, scraped in MeOH:$H_2O$ (1:1). Chloroform was added prior to vortexing, centrifuging, and collecting the aqueous layer. The aqueous layer was dried in a Savant SpeedVac (Thermo Fisher Scientific) without heat. Dried samples were resuspended in 60 µl LC-MS $H_2O$, centrifuged at 15,000 × g at 4 °C for 10 min, and supernatant transferred into HPLC vials for LCMS analysis. Samples were kept at 4 °C on the autosampler tray. Glycolytic and pentose-phosphate pathway metabolites were measured using 1260 Infinity (Agilent)-QTRAP5500 (AB Sciex) LC-MS/MS system. Analyte separation was achieved using a Synergi 2.5 µm Hydro-RP 100A LC Column (100 × 2 mm) (Phenomenex) at ambient temperature. The pair of buffers used were 95:5 (v/v) water:acetonitrile containing 10 mM tributylamine and 15 mM acetic acid (Buffer A) and 100% acetonitrile (Buffer B) flowed at 200 µl/min; injection volume was 5 µl. Scheduled MRM acquisition was performed in negative mode (350 °C, –4500 V). Raw data was extracted using MSConvert (*Chambers et al., 2012*) and in-house MATLAB scripts.

## Metabolic flux analysis

LNCaP cells were seeded at a density of $7.5 \times 10^5$ cells per well into Nunclon D multi-dishes with poly-lysine coating (Thermo Fisher Scientific), with or without transfection of siRNAs (siAR, Silencer Select s1539; siPGD, Silencer Select s10394; siSREBF1, ON-TARGETplus 6720). For the $^{13}$C-labelled glucose time-course experiment, cells were cultured for 46 hr before adding fresh media for a further 2 hr and then spiking in 1,2-$^{13}$C$_2$ glucose at a final concentration of 11 mM (1:1 with natural glucose). Incorporation of the labelled glucose was allowed to proceed for 0, 10, 60, 120, 240, 480, or 900 s. This spike-in strategy (as opposed to media exchange) enabled a rapid time course with minimal disruption to glycolytic fluxes. Experiments were stopped by quenching the cells with ice-cold methanol:$H_2O$ (1:1) and placing plates at –20 °C (prior to cell scraping). After completion of the time course, cell slurries in methanol:$H_2O$ were collected by scraping and transferred into microfuge tubes. Samples were identically processed and assayed as described for metabolomics samples, with the exception that MRMs were configured to quantify mass isotopologues of glycolytic and PPP intermediates. Fluxes through the oxidative and non-oxidative branches of PP pathway were estimated using the accumulation/dilution rate of $m_1$ and $m_2$ isotopologues of Ru5P. Assuming steady-state metabolism, the dilution rate (*D*) of Ru5P was calculated using the continuous stirred-tank reactor (CSTR) equation $m\left(t\right) = m_{maximum} \cdot \left(1 - e^{-D \cdot t}\right) + m_{initial}$, with *m* representing Ru5P mass isotopologues $m_1$ and $m_2$ datapoints generated from the time-course experiment. *D* was estimated using a least-squares Monte Carlo fitting script in MATLAB. Since a Monte Carlo procedure was used to simulate dilution rates, empirical p-values were calculated using the equation: p=(*r* + 1)/(*n* + 1) (*Davison and Hinkley, 1997*), where r is the number of instances the null hypothesis ($H_0$: $D - D_{siCON} \geq 0$) is true and n is the number of simulated replicates (n = 1000).

## ROS assays

Cellular ROS levels were measured using CellROX Orange Flow Cytometry Assay Kits (Life Technologies). Briefly, 24 hr post-seeding ($5 \times 10^5$ cells per well, six-well plate), the cells were treated with or without antioxidant (0.5 mM Trolox) and left to incubate for the indicated time (siRNA, 48 hr; S3, 72 hr). Cells were stained with CellROX Orange and SYTOX Red Stain and analysed by Flow Cytometry (10–30,000 cells/sample) using a BD LSRFortessa X20.

## Ex vivo culture of human prostate tumours

PCa tissue was obtained with informed written consent through the Australian Prostate Cancer BioResource from men undergoing radical prostatectomy at St Andrew's Hospital (Adelaide, Australia). Ethical approval for the use of human prostate tumours was obtained from the Ethics Committees of the University of Adelaide (Adelaide, Australia; approval H-2012-016) and St Andrew's Hospital (Adelaide, Australia). All experiments were performed in accordance with the guidelines of the National Health and Medical Research Council (Australia). The 8 mm core of tissue was dissected and prepared for ex vivo culturing as described previously (*Centenera et al., 2012*). Tissues were treated

with 10 µM Enz or 40 µM S3 for 72 hr. At the time of collection, the tissues were preserved in RNAlater (Invitrogen; Thermo Fisher Scientific) or formalin-fixed then paraffin-embedded.

## Evaluation of AR ubiquitylation

LNCaP cells ($1.5 \times 10^6$ cells per 6 cm plate) were treated with indicated concentrations of S3 ±10 µM MG132, or 10 µM MG132 alone, for 24 hr. Cells were lysed in RIPA lysis buffer. After centrifugation for 10 min at 16,000 g, supernatants were incubated with 0.2 µg anti-AR antibody (Santa Cruz Biotechnology, sc-7305; RRID:AB_626671) for 16 hr at 4 °C with constant rotation, followed by incubation with 20 µl protein G Sepharose beads (Thermo Fisher) for a further period of 2 hr at 4 °C with constant rotation. Beads were washed twice with RIPA lysis buffer and then resuspended in 2× Laemmli sample buffer before samples were assessed by Western blotting.

## Immunohistochemistry

PDE tissue sections were evaluated for target antigens 6PGD, Ki67, and pS6 (Ser235/236) by IHC as described previously (*Centenera et al., 2012*). The antibodies used are shown in *Supplementary file 2*. An automated staining protocol (U OptiView DAB IHC v6 [v1.00.0136]) using the Ventana BenchMark ULTRA IHC/ISH Staining Module (F Hoffmann-La Roche Ltd, Switzerland) was used for the detection of AR. Quantitative image analysis for AR and pS6 (Ser235/236) was completed using FIJI software (ImageJ; http://fiji.sc/Fiji; version 1.52 p). Briefly, images (obtained from NDP viewer version 2.7.52; Hamamatsu Photonics K.K, Hamamatsu City, Japan) were imported and converted into three panels using the *Colour Deconvolution* plug-in and vector haematoxylin and DAB staining (HDAB) commands. The *Adjust Threshold* plug-in was used on the DAB-only images to measure % area (positivity) and reciprocal intensity (RI). The final DAB intensity values were calculated by subtracting RI from maximal intensity (255) and multiplying by % area (positivity). Values from 20 to 70 images per treatment were measured and RI was kept constant for each patient.

## Statistical analysis

Data are displayed as the mean; error bars are standard error. Differences between groups were determined using GraphPad Prism with t tests or one-way ANOVA (with Tukey or Dunnett's post hoc tests), as indicated in the figure legends. A p-value ≤ 0.05 was considered statistically significant.

## Acknowledgements

We acknowledge expert assistance from Deanna Miller, Kayla Bremert, Madison Helm, Samira Khabbazi, Scott Townley, Zeyad Nassar, Elizabeth Nguyen, Shadrack M Mutuku, Courtney Moore, Mark Van der Hoek, Randall Grose, Bianca Varney, and Michelle van Geldermalsen. RNA-seq was performed at the South Australian Health and Medical Research Institute (SAHMRI) Genomics Facility. Flow cytometry analysis was performed at SAHMRI in the ACRF Cellular Imaging and Cytometry Core Facility, which is generously supported by the Australian Cancer Research Foundation, Detmold Hoopman Group and Australian Government through the Zero Childhood Cancer Program. Metabolomics was facilitated by access to Sydney Mass Spectrometry, a core research facility at the University of Sydney. LNCaP-V16D and LNCaP-MR49F cells were a kind gift from Dr. Amina Zoubeidi (Vancouver Prostate Centre, Vancouver, Canada). We are grateful to the study participants, as well as the urologists, nurses, and histopathologists who assisted in the recruitment and collection of patient information and pathology reports through the Australian Prostate Cancer BioResource.

This work was supported by a grant from Cancer Australia (ID 1138766 to LMB, DJL, MMC, IGM). The research programmes of LMB and LAS are supported by the Movember Foundation and the Prostate Cancer Foundation of Australia through a Movember Revolutionary Team Award. LMB and LAS are supported by Principal Cancer Research Fellowships awarded by Cancer Council's Beat Cancer project on behalf of its donors, the State Government through the Department of Health and the Australian Government through the Medical Research Future Fund (PRF1117 and PRF2919).

# Additional information

## Funding

| Funder | Grant reference number | Author |
|---|---|---|
| Cancer Australia | 1138766 | Margaret M Centenera<br>Ian G Mills<br>David J Lynn<br>Lisa M Butler |
| Movember Foundation | MRTA3 | Andrew J Hoy<br>Margaret M Centenera<br>Luke A Selth<br>Lisa M Butler |
| Prostate Cancer Foundation of Australia | MRTA3 | Andrew J Hoy<br>Margaret M Centenera<br>Luke A Selth<br>Lisa M Butler |
| Cancer Council South Australia | Principal Cancer Research Fellowships | Luke A Selth<br>Lisa M Butler |

The funders had no role in study design, data collection and interpretation, or the decision to submit the work for publication.

## Author contributions

Joanna L Gillis, Conceptualization, Data curation, Funding acquisition, Investigation, Methodology, Project administration, Resources, Supervision, Validation, Visualization, Writing – original draft, Writing – review and editing; Josephine A Hinneh, Conceptualization, Data curation, Investigation, Methodology, Supervision, Validation, Visualization, Writing – original draft, Writing – review and editing; Natalie K Ryan, Adrienne R Hanson, Jianling Xie, Investigation, Methodology, Validation, Writing – review and editing; Swati Irani, Data curation, Investigation, Methodology, Writing – review and editing; Max Moldovan, Data curation, Formal analysis, Investigation, Methodology, Visualization, Writing – review and editing; Lake-Ee Quek, Raj K Shrestha, Data curation, Formal analysis, Investigation, Methodology, Validation, Visualization, Writing – review and editing; Andrew J Hoy, Jeff Holst, Funding acquisition, Investigation, Methodology, Supervision, Validation, Writing – review and editing; Margaret M Centenera, Conceptualization, Data curation, Formal analysis, Funding acquisition, Investigation, Methodology, Supervision, Visualization, Writing – review and editing; Ian G Mills, Luke A Selth, Conceptualization, Data curation, Formal analysis, Funding acquisition, Investigation, Methodology, Supervision, Visualization, Writing – original draft, Writing – review and editing; David J Lynn, Lisa M Butler, Conceptualization, Data curation, Formal analysis, Funding acquisition, Investigation, Methodology, Project administration, Resources, Supervision, Visualization, Writing – original draft, Writing – review and editing

## Author ORCIDs

Andrew J Hoy http://orcid.org/0000-0003-3922-1137
Jeff Holst http://orcid.org/0000-0002-0377-9318
Luke A Selth http://orcid.org/0000-0002-4686-1418
Lisa M Butler http://orcid.org/0000-0003-2698-3220

## Ethics

Human subjects: Prostate cancer tissue was obtained with informed written consent through the Australian Prostate Cancer BioResource from men undergoing radical prostatectomy at St Andrew's Hospital (Adelaide, Australia). Ethical approval for the use of human prostate tumours was obtained from the Ethics Committees of the University of Adelaide (Adelaide, Australia) and St Andrew's Hospital (Adelaide, Australia). All experiments were performed in accordance with the guidelines of the National Health and Medical Research Council (Australia).

## Decision letter and Author response

Decision letter https://doi.org/10.7554/eLife.62592.sa1
Author response https://doi.org/10.7554/eLife.62592.sa2

# Additional files

## Supplementary files
- Supplementary file 1. Primers used for quantitative reverse transcription PCR (qRT-PCR).

- Supplementary file 2. Antibodies (primary and secondary) used for western blotting and immunohistochemistry (IHC).
- Transparent reporting form
- Source data 1. Western blot source data, including all individual Western blot images and a summary file.

## Data availability
All data generated or analysed during this study are included in the manuscript and supporting files. Source data files have been provided for Figure 1. Sequencing data have been deposited in GEO under accession code GSE152254.

The following dataset was generated:

| Author(s) | Year | Dataset title | Dataset URL | Database and Identifier |
|---|---|---|---|---|
| Gillis JL, Hinneh JA, Ryan NK, Irani S, Moldovan M, Quek LE, Hoy AJ, Holst J, Centenera MM, Mills IG, Lynn DJ, Selth LA, Butler LM | 2020 | A feedback loop between the androgen receptor and 6-phosphogluoconate dehydrogenase (6PGD) drives prostate cancer growth | https://www.ncbi.nlm.nih.gov/geo/query/acc.cgi?acc=GSE152254 | NCBI Gene Expression Omnibus, GSE152254 |

The following previously published datasets were used:

| Author(s) | Year | Dataset title | Dataset URL | Database and Identifier |
|---|---|---|---|---|
| Array | 2015 | The Cancer Genome Atlas Prostate Adenocarcinoma (TCGA-PRAD) | https://portal.gdc.cancer.gov/projects/TCGA-PRAD | National Cancer Institute, TCGA-PRAD |
| Rajan P, Sudbery IM, Villasevil ME, Mui E, Fleming J, Davis M, Ahmad I, Edwards J, Sansom OJ, Sims D, Ponting CP, Heger A, McMenemin RM, Pedley ID, Leung HY | 2013 | The Wnt/β-catenin-signaling pathway is modulated by androgen ablation therapy for advanced clinical prostate cancer and contributes to androgen independent cell growth | https://www.ncbi.nlm.nih.gov/geo/query/acc.cgi?acc=GSE48403 | NCBI Gene Expression Omnibus, GSE48403 |
| Pomerantz MM, Li F, Takeda D, Chonkar A, Chabot M, Li Q, Cejas P, Vazquez F, Shivdasani RA, Seo J, Bowden M, Lis R, Hahn WC, Kantoff PW, Brown M, Loda M, Long HW, Freedman ML | 2015 | Androgen receptor programming in human tissue implicates HOXB13 in prostate pathogenesis [ChIP-Seq] | https://www.ncbi.nlm.nih.gov/geo/query/acc.cgi?acc=GSE56288 | NCBI Gene Expression Omnibus, GSE56288(GSM1358397) |

*Continued on next page*

*Continued*

| Author(s) | Year | Dataset title | Dataset URL | Database and Identifier |
|---|---|---|---|---|
| Asangani IA, Dommeti VL, Wang X, Malik R, Cieslik M, Yang R, Escara-Wilke J, Wilder-Romans K, Dhanireddy S, Engelke C, Iyer MK, Jing X, Wu YM, Cao X, Qin XS, Wang S, Feng FY, Chinnaiyan AM | 2014 | Therapeutic targeting of BET bromodomain proteins in castration-resistant prostate cancer [ChIP-Seq] | https://www.ncbi.nlm.nih.gov/geo/query/acc.cgi?acc=GSE55062 | NCBI Gene Expression Omnibus, GSE56288(GSM1328950) |
| Barfeld SJ, Urbanucci A, Fazli L, Rennie PS, Yegnasubramanian V, de Marzo AM, Thiede B, Harri IM, Hicks JL, Mills IG | 2017 | Overexpression of c-Myc antagonises transcriptional output of the androgen receptor in prostate cancer [ChIP-Seq] | https://www.ncbi.nlm.nih.gov/geo/query/acc.cgi?acc=GSE73994 | NCBI Gene Expression Omnibus, GSE73994(GSM1907200) |
| ENCODE Project Consortium | 2016 | SREBF1 ChIP-seq on human MCF-7 | https://www.ncbi.nlm.nih.gov/geo/query/acc.cgi?acc=GSE91561 | NCBI Gene Expression Omnibus, GSE91561(ENCFF911YFI) |
| ENCODE Project Consortium | 2011 | SREBF1 ChIP-seq on human HepG2 treated with insulin | https://www.ncbi.nlm.nih.gov/geo/query/acc.cgi?acc=GSM935627 | NCBI Gene Expression Omnibus, GSE31477(GSM935627;ENCFF000XXR) |
| Latonen L, Afyounian E, Aapola U, Annala M, Kivinummi KK, Tammela TTL, Beuerman RW, Uusitalo H, Nykter M, Visakorpi T | 2018 | Prostate cancer study. Study has three groups, bening prostate cancer (BPH), prostate cancer (PC) and castration resistant prostate cancer (CRPC). Analyzed samples were frozen tissuen cut samples. | http://www.peptideatlas.org/PASS/PASS01126 | Peptide Atlas repository, PASS01126 |

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
