## [Decision Letter]

**Acceptance summary:**

This article reports on the regulatory loop between androgen receptor and 6-phosphogluconate dehydrogenase which is a rate-limiting enzyme in the pentose phosphate pathway. Evidence is provided that modulation of 6-phosphogluconate dehydrogenase by androgen receptor is mediated by sterol regulatory element binding protein 1. Moreover, 6-phosphogluconate dehydrogenase appears to be essential for the survival of prostate cancer cells while its expression is elevated in prostate cancer patient specimens. Overall, it was thought that this study is of broad interest inasmuch as it highlights previously unappreciated role of androgen receptor in metabolic reprogramming in neoplasia of the prostate, thus suggesting potentially targetable metabolic vulnerabilities.

**Decision letter after peer review:**

Thank you for submitting your article "A feedback loop between the androgen receptor and 6-phosphogluoconate dehydrogenase (6PGD) drives prostate cancer growth" for consideration by *eLife*. Your article has been reviewed by 3 peer reviewers, including Ivan Topisirovic as Reviewing Editor and Reviewer #1, and the evaluation has been overseen by Erica Golemis as the Senior Editor. The following individual involved in review of your submission has agreed to reveal their identity: Luke Gaughan (Reviewer #3).

The reviewers have discussed the reviews with one another and the Reviewing Editor has drafted this decision to help you prepare a revised submission.

As the editors have judged that your manuscript is of interest, but as described below that additional experiments are required before it is published, we would like to draw your attention to changes in our revision policy that we have made in response to COVID-19 (https://elifesciences.org/articles/57162). First, because many researchers have temporarily lost access to the labs, we will give authors as much time as they need to submit revised manuscripts. We are also offering, if you choose, to post the manuscript to bioRxiv (if it is not already there) along with this decision letter and a formal designation that the manuscript is "in revision at eLife". Please let us know if you would like to pursue this option. (If your work is more suitable for medRxiv, you will need to post the preprint yourself, as the mechanisms for us to do so are still in development.)

Summary:

In this article, Gillis et al. provide evidence suggesting that androgen receptor (AR) targets 6-phosphogluconate dehydrogenase (6PGD), a key enzyme in the pentose phosphate pathway (PPP). In prostate cancer cells, 6PGD levels decrease upon AR depletion but not after enzalutamide treatment. The authors further show that this AR-mediated control of 6PGD levels is regulated by SREBP1. 6PGD depletion or pharmacologic inhibition induce prostate cancer cell death, likely by increasing ROS. Moreover, the authors report that pharmacological inhibition of 6PGD coincides with AR downregulation, thus suggesting an AR/SREBP1/6PGD feedforward loop. 6PGD inhibition also appears to induce AMPK, thereby suppressing mTOR signaling and acetyl-CoA carboxylase (ACC1). Finally, the authors provide evidence that the combination of 6PGD inhibitors with AR antagonists may provide a stronger anti-neoplastic response, thereby highlighting the PPP as a potentially targetable metabolic vulnerability in prostate cancer. This is further corroborated by the findings that 6PGD is upregulated in prostate cancer patient samples. Indeed, the authors demonstrate that 6PGD inhibition impacts cell proliferation, mTOR signaling and AR in patient derived explants (PDEs). Overall, the reviewers agreed the article was of broad interest, inasmuch as it provides a hitherto unappreciated link between AR and potentially targetable metabolic reprogramming in prostate cancer. Nonetheless, several issues pertaining to insufficient understanding of the mechanistic underpinnings of observed phenomena were noted.

Essential revisions:

1. Although previous literature show this in breast and liver cancer cell lines, the authors should conduct ChIP-qPCR in the cell lines they are studying (LNCaP and VCaP) to confirm that SREBP1 is associated with the 6PGD promoter in prostate cancer. If SREBP1 binding to this promoter is observed, does it increase as a result of AR induction and/or decrease with AR inhibition?

2. More mechanistic evidence regarding how AR controls 6PGD levels and subsequently, how 6PGD impacts on AR abundance would improve the study. The authors show that an SREBP inhibitor affects 6PGD levels, but considering potential issues with specificity of inhibitors, SREBP depletion by CRISPR or siRNA should be tested for the ability to attenuate DHT-induced 6PGD expression. Similarly, the experiments using 6PGD inhibitors (Figures 4 and 5) should be complemented by orthogonal genetic approaches.

3. It is not clear how and at which level 6PGD may affect AR levels. Experiments using transcriptional, translational and proteasome inhibitors to address the level at which 6PGD regulates AR are required to strengthen this part of the study. As a related point, the authors should investigate whether overexpression of 6PCG lead to an increase in AR levels.

4. It remains unclear whether the effects of AR on cell survival are dependent on 6PGD. Would enforced expression of 6PCG in the context of AR depletion improve cell viability through engagement of the PPP?

5. A deeper investigation of the effects on PPP is warranted. Although the levels of 6-PG (the substrate of the reaction) have been measured (Figure 3C), levels of the PPP intermediates/products (ribulose-5-phosphate or NADPH) or a final product of the pathway (e.g. ribose-5-phosphate) should be included. Isotopic tracing of 13-C glucose through the PPP following modulation of AR/SEBP1/6PGD would provide more direct evidence that PPP is indeed being affected.

---

## [Author Response]

Essential revisions:1. Although previous literature show this in breast and liver cancer cell lines, the authors should conduct ChIP-qPCR in the cell lines they are studying (LNCaP and VCaP) to confirm that SREBP1 is associated with the 6PGD promoter in prostate cancer. If SREBP1 binding to this promoter is observed, does it increase as a result of AR induction and/or decrease with AR inhibition?

We have spent some time attempting to perform SREBP1 ChIP but have faced a number of challenges outside of our control. Most notably, the validated ChIP antibody for SREBP1 from Santa Cruz Biotechnology (sc-8984), which was used in the ENCODE ChIP-seq studies mentioned by the Reviewers, is no longer commercially available. This led us to try an alternative antibody (Santa Cruz monoclonal anti-SREBP1 2A4). However, despite optimising our experimental conditions to ensure that we selected timepoints for DHT treatment that robustly activated SREBP1 (Author response image 1), we could not demonstrate robust binding of SREBP1 to known target sites at the *SCD* and *LDLR* genes that we used as positive controls (Author response image 1).

**Author response image 1. sa2fig1:** SREBP1 activation and ChIP PCR analysis for AR/SREBP1 in prostate cancer cells. (**A**) LNCaP cells were cultured in the absence or presence of DHT (10nM) for varying timepoints up to 48 hours. Nuclear and cytoplasmic protein extracts were evaluated for the active (cleaved) form of SREBP1. (**B**) DHT-induced recruitment of AR or SREBP1 to canonical AR/SREBP1 binding sites after 4 or 16 hrs culture.

This could not be explained by a problem with the DHT or our ChIP-qPCR protocol, as we showed the expected 40-60-fold recruitment of AR to the *KLK3* enhancer in the same experiment. Collectively these results are consistent with the SREBP antibody being unsuitable for ChIP and, as mentioned, the ChIP-validated antibody is unfortunately no longer available.

2. More mechanistic evidence regarding how AR controls 6PGD levels and subsequently, how 6PGD impacts on AR abundance would improve the study. The authors show that an SREBP inhibitor affects 6PGD levels, but considering potential issues with specificity of inhibitors, SREBP depletion by CRISPR or siRNA should be tested for the ability to attenuate DHT-induced 6PGD expression.

The Reviewers raise an important point about inhibitor specificity, and we now show that siRNA-mediated SREBP1 knockdown also attenuates the DHT-induced protein levels of 6PGD. These new data further strengthen the association between SREBP1 and PGD expression and have been incorporated into the revised manuscript (new Figure 2D).

Similarly, the experiments using 6PGD inhibitors (Figures 4 and 5) should be complemented by orthogonal genetic approaches.

In response to the Reviewers’ suggestion, we have repeated key functional experiments from Figures 4 and 5 using genetic knockdown via siPGD to block 6PGD signalling, instead of the S3 and Physcion inhibitors. Specifically, we have now performed growth, death and ROS assays in the absence or presence of siPGD in 2 additional cell lines (V16D and MR49F), and shown very similar effects to our results using the inhibitors; namely, reduced cell growth and induction of cell death and ROS production. These new data strengthen our conclusions and have been incorporated into the revised manuscript (new Figures 3A, 3B and 3G). We also now include new Western blot data showing that, as we observed using S3, siPGD leads to inhibition of ACC1 and mTOR signalling (new Figure 5—figure supplement 1.).

3. It is not clear how and at which level 6PGD may affect AR levels. Experiments using transcriptional, translational and proteasome inhibitors to address the level at which 6PGD regulates AR are required to strengthen this part of the study. As a related point, the authors should investigate whether overexpression of 6PCG lead to an increase in AR levels.

As suggested by the Reviewers, we exploited transcriptional (actinomycin D), translational (cycloheximide) and proteasome (MG132) inhibitors in our experiments with the 6PGD inhibitor, S3, and assessed the effects on AR RNA (for actinomycin D) or protein (for cycloheximide and MG132) levels. Actinomycin D and cycloheximide did not influence the effect of S3 on AR. Conversely, our experiments with MG132 provide strong evidence that S3 leads to enhanced turnover of AR by the ubiquitin proteasome system. Specifically, we immunoprecipitated AR from LNCaP cells cultured with S3 in the absence and presence of MG132. Importantly, in the presence of MG132, ubiquitinylated AR accumulated in a dose-responsive manner with S3. Accordingly, we can now confidently conclude that S3 is affecting AR at the level of protein turnover, and these new data have been incorporated into the revised manuscript (new Figure 6D).

To address whether overexpression of 6PGD influences AR protein levels, we generated a LNCaP cell line stably overexpressing 6PGD via lentiviral transduction. Over multiple passages in culture, the 6PGD overexpressing line maintained similar AR levels to the control vector-transduced line (Author response image 2). This finding provides further evidence that 6PGD is affecting AR protein stability rather than its protein synthesis.

**Author response image 2. sa2fig2:** Influence of ectopic 6PGD overexpression on AR protein levels in stably transduced LNCaP cells at passages 6/9.

4. It remains unclear whether the effects of AR on cell survival are dependent on 6PGD. Would enforced expression of 6PCG in the context of AR depletion improve cell viability through engagement of the PPP?

Thank you for this suggestion; to address the dependency of AR depletion on 6PGD, we used the LNCaP cell line stably overexpressing 6PGD (Author response image 2) and compared its sensitivity to siAR knockdown with that of the control (empty vector-transduced) line. Author response image 3 shows that both the control and the 6PGD-overexpressing lines were equally sensitive to siAR, indicating that 6PGD alone is not responsible for the growth-inhibitory effects of AR depletion. It is, however, likely that while inhibition or knockdown of 6PGD can block the PPP to influence cell viability, selective overexpression of a single member of the PPP may not sufficiently activate this pathway in a setting where one or more other pathway factors may be limiting. Additionally, AR has many functions in prostate cancer growth beyond regulation of the PPP, and it is therefore not surprising that over-expression of 6PGD alone could not rescue loss of AR.

**Author response image 3. sa2fig3:** Effect of 6PGD overexpression on response of LNCaP prostate cancer cells to siAR treatment. Cells stably transduced with control or 6PGD-overexpressing vectors were transfected with siCon or siAR and viable cell number measured after 6 days in culture. **** P<0.001 c.f. siCon.

5. A deeper investigation of the effects on PPP is warranted. Although the levels of 6-PG (the substrate of the reaction) have been measured (Figure 3C), levels of the PPP intermediates/products (ribulose-5-phosphate or NADPH) or a final product of the pathway (e.g. ribose-5-phosphate) should be included. Isotopic tracing of 13-C glucose through the PPP following modulation of AR/SEBP1/6PGD would provide more direct evidence that PPP is indeed being affected.

We agree with the need to more directly relate our effects on PGD modulation to PPP activity in prostate cancer cells. Accordingly, we have followed the suggestion to perform isotopic tracing with 1,2-^13^C glucose after knockdown of AR/SREBP1/6PGD. PPP flux was estimated over a period of 15 minutes by measuring the incorporation of ^13^C into the immediate product of 6PGD catalytic activity, ribulose-5-phosphate (Ru-5-P). Our new data conclusively show that flux through the oxidative (irreversible) branch of the PPP (i.e. through 6PGD) significantly decreased with knockdown of 6PGD, AR or SREBP1 (included in the revised manuscript as Figures 3D-G, and Figure 3—figure supplement 2C). Interestingly, knockdown of AR and SREBP1 (but not 6PGD) also had a significant impact on flux through the non-oxidative (reversible) branch of the PPP, as determined by evaluating m2 (doubly labelled) Ru5P production via F6P/GAP (Figures 3F-G). Collectively, these glucose tracing data show that targeting 6PGD significantly suppresses PPP activity through the oxidative pathway, an effect that is also evident when targeting the upstream signalling factors AR and SREBP1. We thank the Reviewers for this suggestion, which has markedly strengthened our manuscript’s conclusions.